# A switch point in the molecular chaperone Hsp90 responding to client interaction

Daniel Andreas Rutz [1], Qi Luo[1,2], Lee Freiburger[1,3], Tobias Madl [3,4], Ville R.I. Kaila[1], Michael Sattler[1,3] & Johannes Buchner [1]

Heat shock protein 90 (Hsp90) is a dimeric molecular chaperone that undergoes large conformational changes during its functional cycle. It has been established that conformational switch points exist in the N-terminal (Hsp90-N) and C-terminal (Hsp90-C) domains of Hsp90, however information for switch points in the large middle-domain (Hsp90-M) is scarce. Here we report on a tryptophan residue in Hsp90-M as a new type of switch point. Our study shows that this conserved tryptophan senses the interaction of Hsp90 with a stringent client protein and transfers this information via a cation–π interaction with a neighboring lysine. Mutations at this position hamper the communication between domains and the ability of a client protein to affect the Hsp90 cycle. The residue thus allows Hsp90 to transmit information on the binding of a client from Hsp90-M to Hsp90-N which is important for progression of the conformational cycle and the efficient processing of client proteins.

[1] Center for integrated protein science at the Department Chemie of the Technische Universität München, 84748 Garching, Germany. [2] Soft Matter Research Center and Department of Chemistry, Zhejiang University, Hangzhou 310027, P.R. China. [3] Institute of Structural Biology, Helmholtz Zentrum München, 85764 Neuherberg, Germany. [4] Gottfried Schatz Research Center for Cell Signaling, Metabolism and Aging, Molecular Biology and Biochemistry, Medical University of Graz, 8010 Graz, Austria. Correspondence and requests for materials should be addressed to J.B. (email: johannes.buchner@tum.de)

Hsp90 is the most complex chaperone machinery in the cytosol of eukaryotic cells. It assists the folding and activation of hundreds of clients proteins[1–4], and has also been implicated in general protein quality control, degradation and chromatin remodeling[5,6]. Together with protein kinases and E3 ligases, members of the steroid hormone receptors family represent the most stringent client proteins[7]. Especially, the glucocorticoid receptor (GR) and progesterone receptor (PR) served as models to elucidate the composition of the Hsp90 machinery, including the identification of many co-chaperones involved in the process[8,9]. Several steps in the conformational cycle are targeted by specific co-chaperones which either inhibit or accelerate conformational transitions[10]. Some of these co-chaperones affect the conformational transitions connected to the ATPase cycle and client interactions[11]. Dimerization of Hsp90 occurs via its C-terminal domains (Hsp90-C)[12,13]. In the apo-state Hsp90 is predominantly in an N-terminally open conformation. ATP binding to the N-terminal domain (Hsp90-N) induces local structural rearrangements and repositioning of specific elements that trigger dimerization of the Hsp90-N domains[13–15]. Once Hsp90-N and middle domains (Hsp90-M) are associated, ATP hydrolysis can occur[16]. These conformational changes are slow and rate-limiting for the functional cycle and thus serve to regulate the activity of Hsp90, which has been also observed for other chaperones[17,18].

Post-translational modifications, such as phosphorylation, nitrosylation, SUMOylation and acetylation[2,19] have either local effects, e.g., on the co-chaperone interactions[20–22] or they exert long-range effects as switch points for conformational change[23]. For example, phosphorylation or nitrosylation of residues in Hsp90-C affect the ATPase located in Hsp90-N[24,25]. Molecular dynamics (MD) simulations of Hsp90 combined with biochemical experiments identified further allosteric effects of specific residues[26]. Together, these data show that local changes in side-chain character can have long-range effects on sites distant in the protein. In consequence, Hsp90 is a tightly controlled allosteric molecular machine. However, despite the progress in identifying allosteric switch points in Hsp90, we are still far from being able to reconstruct the network for transducing conformational information across the Hsp90 molecule. Especially information on Hsp90-M, which is important for client binding and co-chaperone interactions, is missing[27–30].

Here, we identified and characterized an important switch point, a conserved tryptophan (W300 in yeast Hsp90) which is localized in Hsp90-M adjacent to the GR client binding site[27]. This residue is exposed to solvent, which is unusual for tryptophan residues due to the hydrophobic properties of the indole moiety and has therefore attracted attention. It has been shown that mutation of this tryptophan to alanine (W300A) impairs yeast growth at 30 °C, affects client processing and co-chaperone binding[31–35]. These results suggest that W300 plays a direct role concerning interactions with the M-domain of Hsp90. Here, we address the function of W300 in vivo, in vitro and in silico to probe whether it functions locally within the interaction surface of Hsp90-M or whether it has a general regulatory function in the context of the conformational changes associated with client binding and activation. We determine the functional defects of W300 mutants and provide structural explanation. Our results demonstrate that W300 is a molecular switch point which responds to client interaction and induces long-range conformational changes in Hsp90 required for client activation.

## Results

**Tryptophan 300 is essential for viability and GR maturation.** The tryptophan residue at position 300 (W300) in S. cerevisiae

Hsp90 (Hsp82) is evolutionary conserved from yeast to man as seen in the sequence alignment of different Hsp90 homologs (Fig. 1a). Interestingly, the residue is located close to the GR client-binding site, in a strongly surface-exposed unstructured loop as seen in the crystal structure of the closed Hsp90 (Fig. 1a)[14,27]. We find that the substitution of this tryptophan by alanine results in a severe growth defect of S. cerevisiae at 30 °C (Fig. 1b), in agreement with previous data[31,32]. When we substitute the tryptophan by a glutamate or lysine residue (W300E/K), and introduce the variants as the sole source of Hsp90 into yeast, the variants are not viable, pointing towards an important role of this residue in Hsp90 function (Fig. 1b). In stark contrast to W300, mutation of a second tryptophan in Hsp90-M, W277 which is even more conserved, to alanine results in a viable mutant with no growth defects (Fig. 1a, b). W277 is located in the hydrophobic core of Hsp90-M[14] and one could thus have expected larger effects on the biological activity of Hsp90 (Fig. 1a, b).

To address the importance of having an aromatic residue at position 300, we generated substitutions against phenylalanine and tyrosine (W300F/Y). These variants are able to rescue viability, underlining that the in vivo Hsp90 function depends on an aromatic residue at this position (Fig. 1b). To gain further insight into the functionality of the tryptophan residues, we tested the ability of the viable mutants (W300A/F/Y and W277A) to mature the stringent client protein GR in vivo using a reporter assay[36]. The aromatic substitutions, W300F and W300Y, as well as W277A, support GR activation (Fig. 1c). We observed a two-fold increase in GR activity as compared to wild-type (wt) Hsp90, indicating that these mutants are slightly more efficient in GR-maturation. In contrast to these activating mutations, W300A showed a reduction of about 50% in GR activation in comparison to wt Hsp90 (Fig. 1c) consistent with previous results[31] and implying an important role of the tryptophan moiety in the client processing by Hsp90. Of note, these effects were not due to differential expression of the mutants as all viable variants showed comparable protein levels in a western blot analysis (Supplementary Fig. 1A).

**W300 is affected by GR-LBD binding to Hsp90-M.** Previously, we employed paramagnetic relaxation enhancement (PRE) experiments using PROXYL-labeled GR-ligand binding domain (GR-LBD) and the [15]N-labeled Hsp90-M domain to map the GR-LBD binding site on Hsp90[27]. These experiments suggested that several residues are involved in the client-binding site, including a previously unassigned tryptophan. To probe the identity of this tryptophan, we recorded [1]H, [15]N-HSQC NMR spectra of wt Hsp90-M and W300A Hsp90-M. Comparison of the spectra allowed us to unambiguously assign W300 as the involved residue, since the typically downfield shifted signal of the tryptophan indole moiety disappears in the W300A Hsp90-M spectrum (Fig. 1d). The NMR spectra of wt and W300A Hsp90-M are highly similar indicating the absence of significant alterations in the structure. However, in the W300A mutant changes in amide chemical shifts or intensities are observed for a few residues near W300 (I296, L304;) but also for residues far away from W300 (K388, I505, Q512, I524;) (Supplementary Fig. 1, Supplementary Table 1). This suggests that W300 also influences distal regions. In our previous experiments, this tryptophan was strongly affected by a spin-label on the GR-LBD. This suggests on the one hand that the spin-label must be in close proximity to W300 as it strongly reduces the indole-signal intensity. On the other hand, W300 is not likely within the core-binding site, as the spin-label would then lead to an impaired binding of the GR-LBD which was not the case.

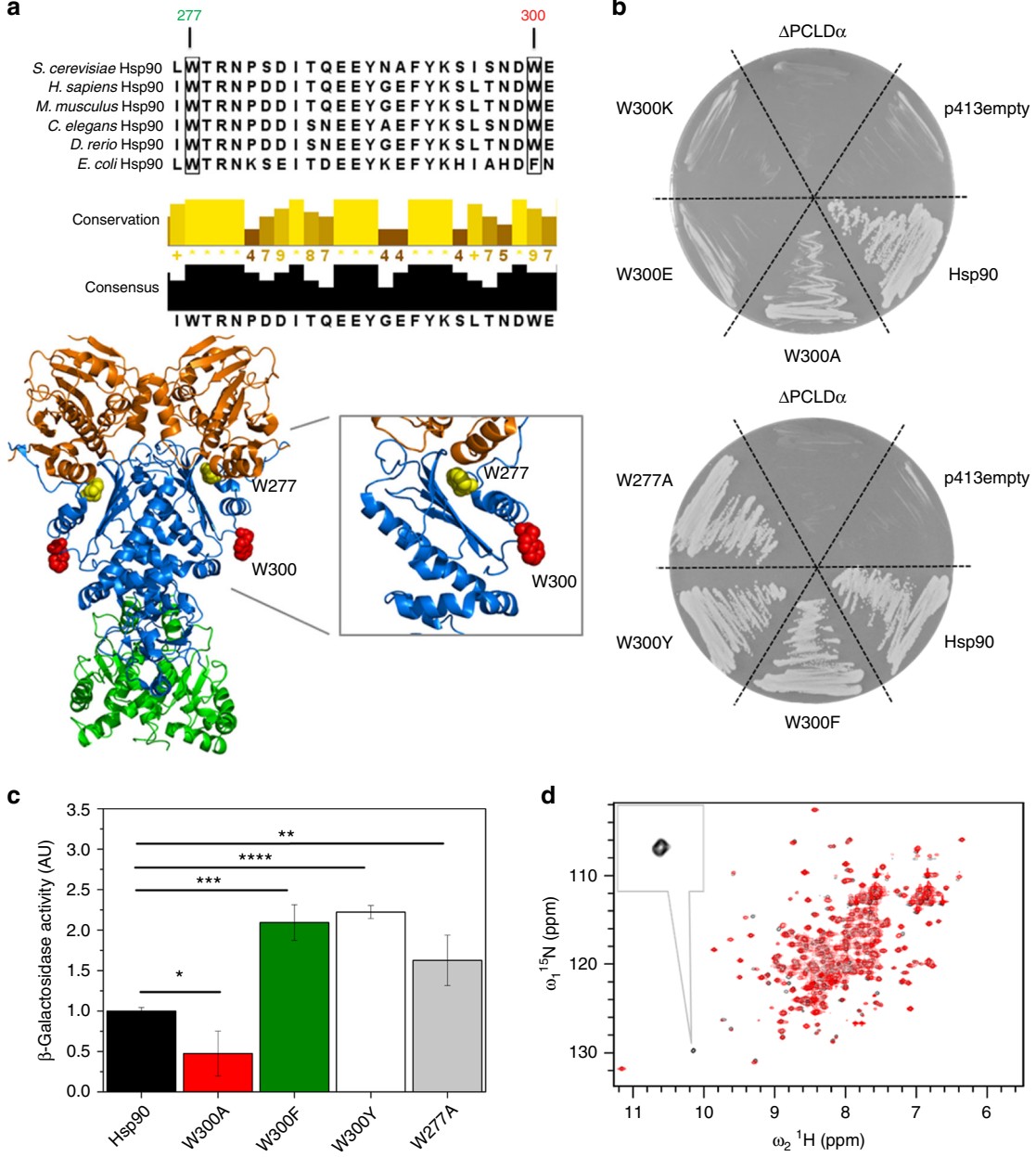

**Fig. 1** The evolutionary conserved W300 affects yeast viability and GR maturation in vivo. **a** Sequence alignment of different Hsp90 homologs (UniProt entries: P02829 (yeast Hsp82), P08238 (human Hsp90β), P11499 (murine Hsp90β), Q18688 (*C. elegans* Hsp90), O57521 (zebrafish Hsp90β), P0A6Z3 (HtpG, *E. coli* Hsp90)) and crystal structure of yeast Hsp90 (PDB ID: 2CG9); Hsp90-N (1–258) is colored in orange, Hsp90-M (259–527) in blue, Hsp90-C (528–709) in green, W300 in red and W277 in yellow (see label); **b** Viability of yeast containing Hsp90 mutants as the sole source of Hsp90. Yeast were grown on respective selective media including 5′FOA. The shuffling strain ΔPCLDα and ΔPCLDα transformed with the empty p413-GPD vector were included as negative controls. ΔPCLDα transformed with p413-GPD containing wt Hsp90 served as positive control. Mutations in Hsp90 are as indicated. **c** GR-maturation by Hsp90 mutants in vivo. ß-galactosidase reporter assay in yeast; plotted are ß-galactosidase activities for the wt and respective Hsp90 mutants. Activities were normalized to ß-galactosidase activity in the presence of wt Hsp90. The mean values and standard deviations from three biological replicates are indicated ($n = 3$). Statistical significance was assessed using a two-sample *t*-test and a level of significance of 0.05. **d** $^1$H, $^{15}$N HSQC experiments for Hsp90-M (black) and W300A Hsp90-M (red); the inset shows the downfield shifted tryptophan signal of interest at greater magnification

Altogether, this is consistent with the idea that W300 is not directly involved in client binding but has a regulatory role in the client interaction.

**W300 mutants are affected in activity and interactions.** To gain further insight in the contribution of W300 to the mechanism of Hsp90, we performed in vitro experiments with the mutant proteins purified from *E. coli*. Thus, post-translational modifications were absent. All mutants except W277A show similar stabilities in thermal unfolding experiments (Supplementary Fig. 2, Table 1) indicating that their structures are not significantly affected by the mutations. In contrast, the mutation at position 277 results in a reduction of the melting temperature by 4 °C consistent with a perturbation of the hydrophobic core (Supplementary Fig. 2, Table 1).

**Table 1 Summarized data for the W300/W277 Hsp90 mutants**

| Hsp90 | wt | W300A | W300E | W300K | W300F | W300Y | W277A |
|---|---|---|---|---|---|---|---|
| $T_M$ (°C) | 60.7 | 60.6 | 60.4 | 60.4 | 60.4 | 60.7 | 56.0 |
| $k_{cat}$ (min$^{-1}$) | 0.5 ± 0.0 | 0.9 ± 0.1 | 1.1 ± 0.1 | 0.8 ± 0.0 | 0.6 ± 0.1 | 0.6 ± 0.1 | 0.5 ± 0.0 |
| +GR-LBD (min$^{-1}$) | 0.2 ± 0.0 | 0.9 ± 0.0 | 1.2 ± 0.0 | 0.9 ± 0.1 | 0.7 ± 0.0 | 0.6 ± 0.0 | 0.4 ± 0.0 |
| $t_{1/2\ form}$ (s) | 53.1 ± 4.7 | 85.4 ± 8.5 | 79.0 ± 7.1 | 63.4 ± 9.5 | 34.8 ± 3.6 | 37.5 ± 9.3 | 35.9 ± 12.3 |
| $t_{1/2\ open\ chase}$ (s) | 29.7 ± 2.6 | 43.2 ± 2.1 | 58.9 ± 12.6 | 61.3 ± 6.4 | 67.3 ± 4.0 | 49.4 ± 3.9 | 45.9 ± 5.9 |
| +GR-LBD (s) | 49.3 ± 4.0 | 44.4 ± 4.2 | 55.7 ± 3.6 | 48.0 ± 1.4 | 48.3 ± 12.1 | 48.3 ± 6.9 | 41.2 ± 6.1 |
| $t_{1/2\ close}$ (s) | 77.5 ± 1.2 | 75.6 ± 9.7 | 61.9 ± 5.8 | 75.5 ± 14.0 | 154.1 ± 31.6 | 165.5 ± 10.0 | 497.4 ± 29.9 |
| +GR-LBD (s) | 1165.5 ± 329.2 | 148.0 ± 22.6 | 131.7 ± 9.4 | 161.3 ± 9.4 | 279.1 ± 20.2 | 226.1 ± 22.9 | 1245.5 ± 154.1 |

Melting temperatures ($T_M$) as determined by TSA, ATPase activity ($K_{cat}$) in the absence and presence of the GR-LBD, half-lives for the formation of the FRET hetero-dimer ($T_{1/2\ form}$), half-lives for the chase experiments with the open Hsp90 hetero-dimer in the absence and presence of the GR-LBD ($T_{1/2\ open\ chase}$) and closing half-lives upon the addition of ATPγS in the absence and presence of the GR-LBD ($T_{1/2\ close}$); melting temperatures were calculated using the Boltzmann fit and reaction half-lives using a single-exponential decay fit

To test for functional consequences of the mutations, we performed ATPase assays (Fig. 2a, Table 1). We find that the aromatic variants at position 300 (W300F/Y) and W277A display wt-like activities, whereas a roughly two-fold increased ATP turnover was observed for the glutamate, lysine and alanine mutants. For W300A, this is in line with published data[32]. Previously, we and others had shown that the GR-LBD inhibits the ATPase activity of Hsp90[27,37]. For the W277A mutant, we observed a similar effect (Fig. 2a, Table 1). When we tested the W300 mutants, surprisingly, none of them showed an inhibition of the ATPase upon client binding (Fig. 2a, Table 1). These results support our idea of a regulatory role. W300 may not be directly involved in the interaction but in the regulation of conformational changes required for efficient binding of the GR-LBD. This is supported by the notion that a simple decrease in affinity should be overcome by increasing GR-LBD concentrations. However, even very high concentrations (>30 μM) did not induce a wt-like inhibition of the Hsp90 W300A ATPase activity (Fig. 2b).

**Alterations in the conformational cycle of the W300 mutants**. To test the influence of mutations at position 300 on the conformational cycle of Hsp90, we employed a FRET system in which donor-labeled and acceptor-labeled Hsp90 dimers allow to follow the global conformational changes of the Hsp90-dimer, i.e., closing and reopening of the dimer, as well effects of co-chaperones and client binding on these reactions[17,27]. Here, we used N-terminally ATTO488/550-labeled Hsp90 (D61C) dimers. Changes in the distances within the Hsp90 dimer result in an increased (closer) or decreased (more distant) acceptor fluorescence signal which gives an accurate measure on the global conformational state of the Hsp90 dimer[17]. First, we monitored the formation of the FRET hetero-dimer complex by mixing equimolar amounts of donor-labeled and acceptor-labeled Hsp90 which should result in an increase in the acceptor fluorescence (Supplementary Fig. 3A, Table 1). The kinetics of this reaction depend on the conformational state of the Hsp90 species: more closed or intertwined dimers exchange on longer timescales and form FRET hetero-dimers more slowly than open dimers which are only C-terminally dimerized. The wt protein has a half-life of 53 s for the formation of the hetero-dimer and similar kinetics are observed for W300K. However, the W300A and W300E mutants display longer half-lives indicative of a more closed conformation in comparison to the wt which also correlates with the accelerated ATPase activity of the mutants seen in Fig. 2a. In contrast, W300F and W300Y as well as W277A have shorter half-lives than the wt, implicating a more open conformation (Supplementary Fig. 3A, Table 1).

Next, we tested the influence of the GR-LBD on the conformation of the Hsp90 variants (Supplementary Fig. 3B, Table 1). To determine the effects on the open states, the pre-formed, open (absence of nucleotides) hetero-dimers were incubated in the absence or presence of the GR-LBD, followed by the addition of an excess amount of unlabeled Hsp90 to the FRET complex. The kinetics of the disruption of the open FRET complex depends, similar to the formation of the heterodimers, on the dimerization state of the dimers. We find that the presence of the GR-LBD almost doubles the half-life of the wt hetero-oligomer, consistent with the notion that this client induces a semi-closed conformation in Hsp90 and crosslinks the protomers[27]. For all of the W300 mutants, no differences in the half-times for complex disruption could be observed in the presence of the GR-LBD indicating that W300 is required to form a semi-closed conformation induced by the GR-LBD.

To determine the effects of the mutants on the closing reaction directly, we induced the formation of the N-terminally closed state by the addition of ATPγS which can be detected via an increase in the acceptor fluorescence. Compared to the wt, we find that the closing kinetics are not affected by the alanine, lysine or glutamate substitutions. This indicates that the intrinsic, nucleotide-supported closing reaction is not affected by the mutations, despite its conformational alterations. W300F and Y showed slightly increased half-lives for the closing reaction, whereas for W277A, there is a moderate increase in the half-life, supporting the idea that the mutation induces a more open conformation (Fig. 3a, Table 1). The GR-LBD had been shown to preferentially bind a semi-closed conformation and to slow down the closing kinetics[27]. Surprisingly, all W300 mutants show a strikingly different behavior: the closing reaction is only slightly affected by the addition of the GR-LBD. The half-lives of the reaction exhibit a two-fold increase in the presence of the GR-LBD as compared to a 12-fold increase observed for the wt protein (Fig. 3a, Table 1). In line with the ATPase results, the aromatic substitutions (W300F/Y) do not support the inhibition of the cycle by the GR-LBD, emphasizing the importance of a tryptophan at position 300. Together with the chase experiments in the absence of nucleotides, the results emphasize the importance of W300 in forming a specialized semi-closed conformation induced by the GR-LBD that is susceptible to the cycle inhibiting effects of the client.

To test for the effect of the mutations on the re-opening of the FRET complex in the presence of the GR-LBD, we formed the closed FRET complex in the presence of the GR-LBD and ATPγS and added unlabeled Hsp90 to determine the subunit exchange. The signal amplitudes in these experiments are a good measure for the reopening of the Hsp90 dimer. The larger the decrease in

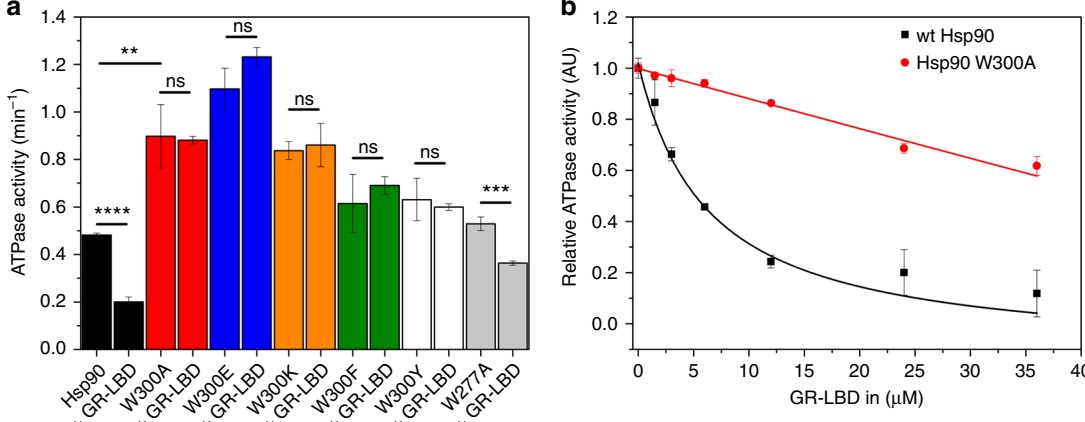

**Fig. 2** ATPase activity of Hsp90 mutants in the presence and absence of the GR-LBD. **a** Absolute ATPase activities of different Hsp90 mutants in the absence and presence of the GR-LBD. ATPase activity for wt Hsp90 is shown in black, for W300A in red, for W300E in blue, for W300K in orange, for W300F in green, for W300Y in white and for W277A in gray. Shown are mean ATPase activities of three independent measurements including the standard deviation represented as black bars ($n = 3$). Statistical significance was assessed using a two-sample $t$-test and a level of significance of 0.05. The color code will be maintained for the mutants throughout the manuscript. **b** ATPase activity of wt Hsp90 in black and W300A in red in the presence of increasing amounts of GR-LBD; mean relative ATPase activities determined from three independent measurements ($n = 3$) were plotted against the GR-LBD concentration. The indicated standard deviations were calculated from three independent measurements

the acceptor signal, the more FRET signal is lost implying an increased population of re-opened Hsp90. In the absence of the GR-LBD, wt Hsp90 and Hsp90 W300A show comparable re-opening as both traces display similar signal amplitudes (Fig. 3b). In the presence of the GR-LBD, the dissociation of the wt Hsp90 FRET complex is partly inhibited, as deduced from a smaller decrease in the acceptor signal (Fig. 3b). This is indicative of the presence of a stabilized, partially closed state which cannot be dissolved by the excess of unlabeled Hsp90. In contrast, we find that the W300A mutant is not sensitive to GR-LBD in the re-opening and the signal amplitude of the re-opening reaction remains virtually unchanged in comparison to the measurement in the absence of GR-LBD (Fig. 3b). Similar results were also obtained for the other mutants (Supplementary Fig. 3C), suggesting that the mutants are not able to form a stabilized, semi-closed conformation.

**Conserved effects on client activation from yeast to man.** Having established the basic influence of W300 on the regulation of Hsp90 conformation, we wanted to test how conserved the effects are. To this end, we generated the respective substitutions (W312A/E/K) in human Hsp90β. These mutants behave similar to the yeast Hsp90 variants when tested for effects on viability in yeast, thermal stability and ATPase activity (Supplementary Fig. 4A–C). In line with the literature, wt human Hsp90β was able to substitute endogenous wt Hsp90 in yeast[38]. In the case of the human Hsp90β W312A mutant, the phenotype is more pronounced than for yeast Hsp90 W300A and thus the yeast are no longer viable (Supplementary Fig. 4A). Similar to the yeast homologs, the ATPase activity of wt human Hsp90β could be inhibited by the GR-LBD while the mutants were not affected in their activity by the presence of the client (Supplementary Fig. 4C). We thus conclude that the human variants show the same behavior as the yeast mutants suggesting that the effects are evolutionary conserved.

Next, we tested how the human mutants affect ligand-binding recovery of the GR-LBD after incubation with Hsp70/Hsp40 and ATP. It had been shown previously that the GR-LBD is partly unfolded and unable to bind its ligand when associated with

Hsp40 and Hsp70[37]. Accordingly, we could not detect hormone binding when the GR-LBD was incubated with Hsp40/70 (Fig. 3c, Supplementary Fig. 4D). When we added the Hsp90 co-chaperones Hop and p23 together with wt Hsp90β, recovery of hormone binding was achieved. Interestingly, the W312 mutants did not support the recovery of ligand binding to the same extent as wt Hsp90 (Fig. 3c, Supplementary Fig. 4D). Ligand binding reached only 40% for W312A and less than 30% for W312E/K as compared to wt Hsp90β, consistent with the population of different Hsp90 conformations for the mutants. Thus, alterations in the conformational cycle lead to a significantly decreased fraction of ligand-bound and reactivated GR-LBD.

**W300 affects conformational steps following client binding.** Our results suggest that W300 plays an important role in triggering the conformation needed to promote the formation of a GR-LBD that is fully competent for hormone-binding and is able to establish its cycle inhibiting effects. To determine the involvement of W300 in the direct binding of the client, we assayed the interaction by fluorescence-coupled AUC using ATTO488-labeled GR-LBD and Hsp90/-mutants (Fig. 3d, Supplementary Fig. 4E). We performed the binding experiments using yeast Hsp90 as it is known that the nucleotide-dependent conformational changes are more significant than in the human system which makes the yeast system more accessible to detect alterations in the GR-LBD binding. In the absence of nucleotide, only slight changes in the amount of complex were detected for all the Hsp90 W300 mutants. Also in the presence of ADP, which supports the open conformation of Hsp90[13], the binding of the GR-LBD was similar to wt Hsp90 or mutants (Fig. 3d). These experiments show that the binding of the GR-LBD to the open state of Hsp90 is not significantly affected by the mutations. Thus, we conclude that the defects in client activation of the mutants must be related to subsequent steps. Therefore, we tested complex formation in the presence of ATP which increases the affinity of Hsp90 for the GR-LBD due to the formation of a semi-closed conformation[27]. For the W300 mutants, we find that the amount of complex formation is similar to that in the absence of ATP. Thus, the response to ATP observed for the wt protein is

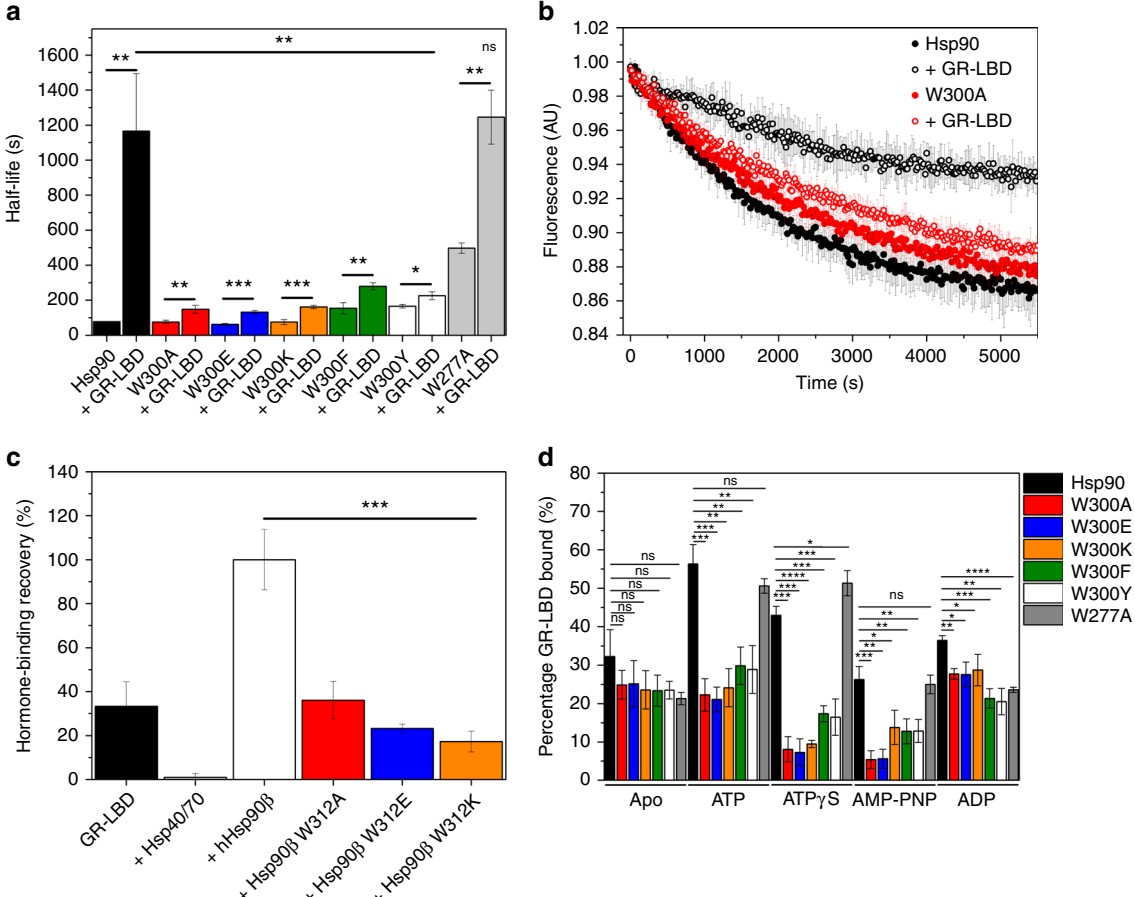

**Fig. 3** Conformational effects of mutations of the conserved tryptophan. **a** Closing reaction of the FRET hetero-dimer; closing of the wt/mutant Hsp90 dimer was monitored after addition of ATPγS in the presence and absence of the GR-LBD. Mean reaction half-lives from three independent measurements were derived from the original traces and plotted ($n = 3$). Standard deviations based on three independent measurements are indicated. Statistical significance was assessed using a two-sample $t$-test and a level of significance of 0.05. **b** Chase experiments with the closed wt/W300A Hsp90 dimer in the absence and presence of the GR-LBD. Chase experiments were initiated by the addition of unlabeled wt Hsp90. The mean fluorescence and standard deviations of three independent measurements were plotted ($n = 3$). **c** Fluorescence anisotropy hormone-binding recovery measurements with human Hsp90β W312 mutants; normalized mean fluorescence anisotropy end values of the binding kinetics were normalized to the value in the presence of the full chaperone system and wt human Hsp90β. The measurements were performed in triplicates ($n = 3$). Error bars indicate standard deviations from three independent measurements. Statistical significance was assessed using a two-sample $t$-test and a level of significance of 0.05. **d** AUC-analysis of GR-LBD binding to Hsp90 W300 mutants in different nucleotide states; bars indicate mean amount of labeled GR-LBD found in the Hsp90/mutant complex in the presence of Hsp90 and indicated nucleotides. Measurements were performed in triplicates to rule out experimental fluctuations ($n = 3$). Error bars represent standard deviations from three independent measurements. Statistical significance was assessed using a two-sample $t$-test and a level of significance of 0.05

abolished. This effect becomes even more pronounced in the presence of the slowly hydrolysable ATP analog ATPγS or the non-hydrolysable analog AMP–PNP which shift the conformational equilibrium towards completely closed Hsp90 conformations[17,27]. Under these conditions, we find only 5% or less GR-LBD in complex especially with Hsp90 W300A and W300E. Taken together, the analysis of complex formation reveals that the W300 mutants are in principle not compromised in their binding to the GR-LBD. Instead, W300 represents an important position for inducing a conformation which allows Hsp90 to respond to client-binding and processing. Of note, the mutant W277A shows wt-like behavior in terms of nucleotide-dependent GR-LBD binding emphasizing the special role of W300.

**Conformational alterations result in reduced client binding**. To acquire structural information on the Hsp90 W300 mutants, SAXS measurements were performed with W300A, W300E and

W300K in the absence of nucleotide and in the presence of ATP, ATPγS or AMP–PNP (Fig. 4 and Supplementary Figs. 5–8). In these experiments, we addressed whether W300 mutations result in altered Hsp90 conformations that would lead to differential binding of the GR–LBD. The SAXS measurements reveal intriguing differences in the conformations of the mutants. In the absence of nucleotide, W300A and W300E exhibit conformations similar to the wt protein, although slightly more compact, as deduced from the radius of gyration ($R_g$) and maximum dimensions ($D_{max}$) (Fig. 4a, Supplementary Tables 2 and 3). Surprisingly, in the absence of nucleotide W300K shows a more elongated conformation in comparison to the wt protein. Addition of ATP leads to a ca. 5% more compact conformation for the W300A and E mutants in comparison to wt Hsp90. Although both proteins show increased compaction, the pair-distance distributions, $P(R)$, are significantly different at shorter distances which might reflect differences in their conformations (Fig. 4a, b, Supplementary Table 3). W300K displays a similar conformation

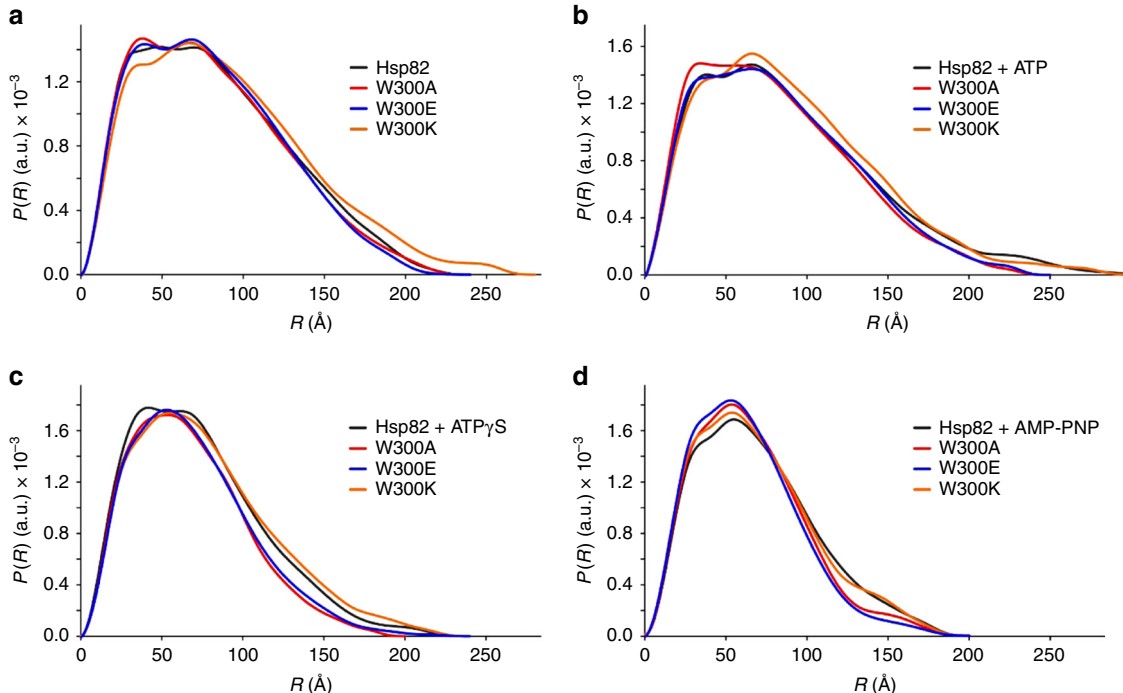

**Fig. 4** SAXS-analysis of wt Hsp90 and Hsp90 W300-mutants. Measurements were performed as single-measurements at different sample concentrations ($n = 1$). $P(R)$-curves of wt and mutant Hsp82 (W300A/E/K) **a** in the absence of nucleotide, and **b** in the presence of 2 mM ATP, **c** ATPγS, or **d** AMP–PNP. Black traces indicate wt Hsp90, red traces Hsp90 W300A, blue traces Hsp90 W300E and orange traces Hsp90 W300K

as the wt protein in the presence of ATP, but with larger differences at shorter distances suggesting that its conformation differs from the wt protein (Fig. 4b). The measurements in the presence of ATPγS and AMP–PNP amplify the effects observed in the presence of ATP (Fig. 4c, d, Supplementary Table 3). Hsp90 W300A and E are more closed than wt Hsp90 ($R_g$ 53.7/55.0 vs. 58.5 Å). The wt protein resides to 40–50% in the closed conformation, whereas W300A and W300E are 60–70% in the closed conformation (Supplementary Fig. 9, Supplementary Table 3). Similar to the measurements in the presence of ATP, W300K is in a similar conformation as wt Hsp90 with slight differences at short distances of the $P(R)$ distribution, implying alterations in the conformation (Fig. 4c, d). The SAXS data implicates that mutation of W300 leads to a shift in the conformational spectrum of Hsp90 to a more closed state, especially for W300A and E and generally altered conformations for W300K in the presence of ATP, ATPγS or AMP–PNP.

**Molecular mechanism of switch point action**. To derive an atomistic picture of the conformational consequences of mutating W300, we performed large-scale MD simulations of the full-length dimeric closed yeast wt Hsp90 (PDB ID: 2CG9) and models for W300A/E/K. In wt Hsp90, W300 forms transient cation–π interactions with K294 which is positioned in the α-helix formed by residues 286–297 directly above the tryptophan as it is located within the range of ca. 5–6 Å (Supplementary Fig. 10), a distance in which most cation–π interactions involving tryptophans occur[39]. This interaction stabilizes the structure of Hsp90-M, and mutations lead to an altered interaction pattern of the W300 loop and the Hsp90-M structure, resulting in conformational changes that propagate also to surrounding helices (Fig. 5, inset). During the 200 ns MD trajectories, all mutant Hsp90 constructs undergo conformational changes that lead to a more compact global structure with increased interactions

between the monomers (Fig. 5a). In the presence of an alanine and glutamate at position 300 (W300A/E), the negatively charged loop (D299, E301, D302) associates significantly stronger with the surrounding positive charge (K294), whereas the mutations to lysine (W300K) conversely lead to a stronger repulsion of K300 and K294 and several contacts with the surrounding carboxyl-groups (D299, E301, D302) (Fig. 5, inset). The loss of the cation–π interaction in the W300 mutations is thus compensated by formation of more compact conformations, decreasing the radius of gyration of Hsp90 (Fig. 5b, Supplementary Fig. 11). For several residues that show chemical shift perturbations or intensity changes, we also observe an increase in the root-mean-square-fluctuations (RMSF) (Supplementary Table 1). Hence, comparable regions in Hsp90-M were affected both experimentally and computationally, supporting the significance of our MD simulation data (Supplementary Fig. 12A, B). Further, these observations complement our SAXS data as MD captures changes in the closed state that are not accessible in the open-closed equilibrium monitored in the SAXS measurements (Fig. 4).

We observe most significant conformational changes in the MD trajectories in the Hsp90-M domain between wt Hsp90 and the mutants (Fig. 5a). To probe whether the W300-mutations could also induce changes in Hsp90-N, we calculated the electrostatic solvation free energy of ATP bound to Hsp90. Interestingly, the ATP molecule seems to become better stabilized in all mutant variants in comparison to wt Hsp90 (Supplementary Fig. 12C, D), suggesting that a stronger Hsp90 dimerization might favor the ATPase activity, as shown experimentally (Fig. 2a, Table 1) and as also suggested in the literature[40]. To investigate possible mechanisms for these long-range coupling effects, we analyzed dynamic networks based on the MD trajectories to identify force propagation pathways between the residue 300 (W/A/K/E) and the Hsp90-N domain. Supplementary Fig. 12C shows the shortest identified force propagation pathway from W300 to the catalytically critical E33 of Hsp90-N, which is

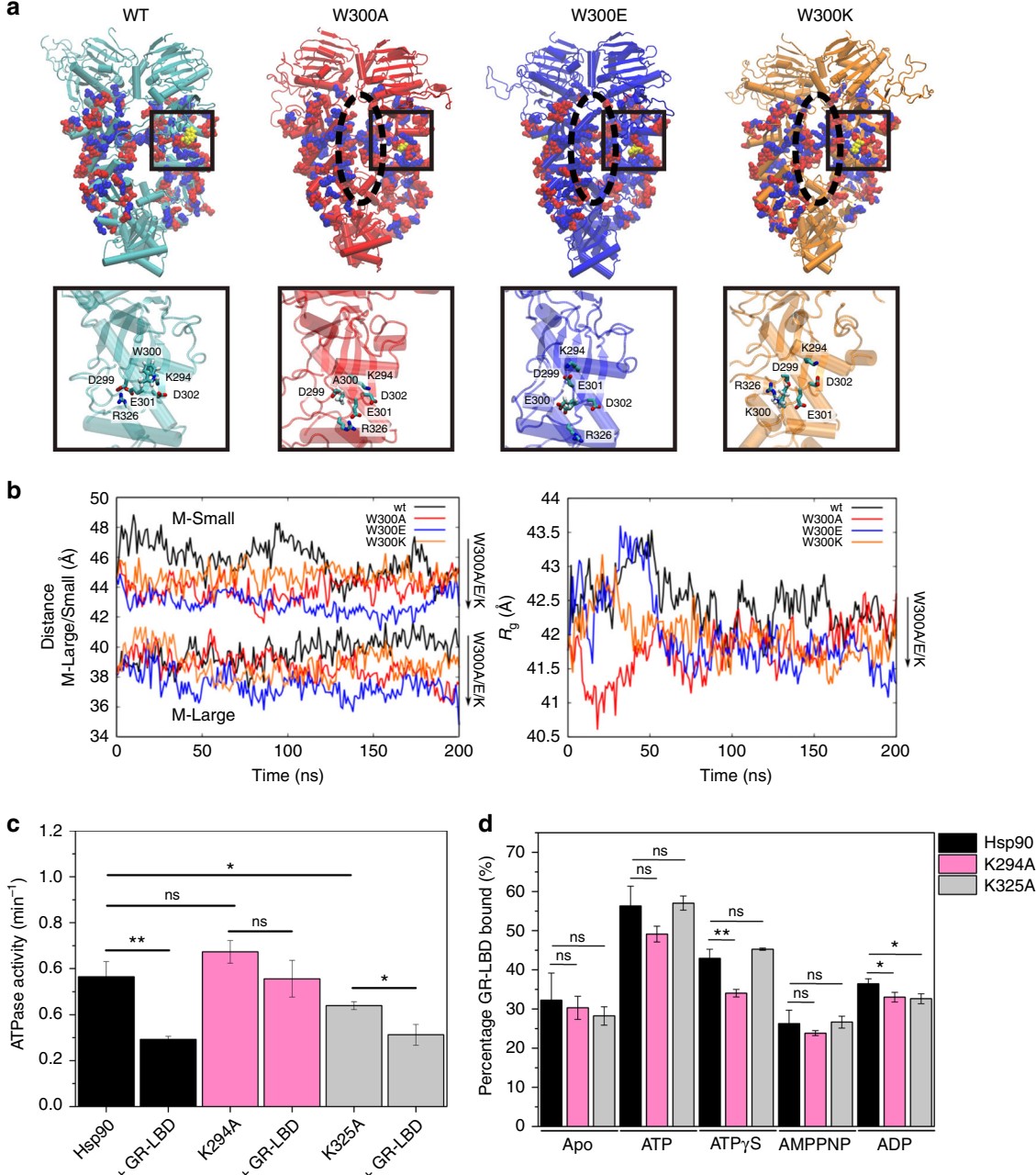

**Fig. 5** Molecular dynamics simulations of the Hsp90 W300 mutants. **a** Global structures of the wt and W300A, W300E, W300K mutants of Hsp90, obtained from 200 ns MD simulations of each model. Charged residues in Hsp90-M are shown in van der Waals representation with negative residues in red, and positive residues in blue. In comparison to wt Hsp90, the mutants have more closed M-domains formed by inter-domain contacts (dotted lines). Inset: central residue interactions near W300: W300 forms transient cation–π interactions with K294, whereas replacements to A/E/K leads to increased repulsion/attraction between charge residues that induce conformational changes in surrounding loops and helices (see text). **b** Distance between the center of mass of two M-Large (res 264–426) domains and two M-Small (res 427–526) domains during the MD trajectories (left), and calculated radius of gyration $R_g$ (right) showing that the W300 mutations lead to a more compact Hsp90. **c** Absolute ATPase activities of Hsp90 K294A and K325A in the absence and presence of the GR-LBD. ATPase activity for wt Hsp90 is shown in black, for K294A in pink and for K325A in light gray. Shown are mean ATPase activities of three independent measurements including the standard deviation represented as black bars ($n = 3$). Statistical significance was assessed using a two-sample $t$-test and a level of significance of 0.05. **d** AUC-analysis of GR-LBD binding to Hsp90 K294A and K325A in different nucleotide states; bars indicate mean amount of labeled GR–LBD found in the Hsp90/mutant complex in the presence of Hsp90 and indicated nucleotides. Measurements were performed in triplicates to rule out experimental fluctuations ($n = 3$). Error bars represent standard deviations from three independent measurements. Statistical significance was assessed using a two-sample $t$-test and a level of significance of 0.05

likely to act as the proton acceptor in ATP hydrolysis[14]. In wt Hsp90, W300 forms transient cation–π interaction with K294 on the smaller α-helix in the Hsp90-M large domain (residues 286–297), as mentioned above, from which the force propagates via a α-helix (residue 286–297), over the neighboring β-sheet

(residues 303–324, 341–345, 365–371) to E33. A similar pathway is also observed in the W300E model, where E300 electrostatically interacts with K294. Upon mutation W300 to alanine, we identify a different force propagation pathway via a loop structure that connects the α-helix and β-sheets instead of K294, which

increases the length of the interaction pathway. Analysis of this structural network further suggest that the mutations lead to local lowering of the overall packing density in this structural network, which may result in the increased formation of ion-pairs between the dimer interface (Supplementary Figs. 10 and 13), which in turn may modulate the ATP activity and closing energetics. Interestingly, only one pathway is found in the mutant W300K, probably due to the repulsion between K300 and K294 that weakens the interaction of the former residue with surrounding residues. The identified force propagation pathways suggest that the local conformational changes induced by the mutations of W300 could trigger the global conformational transitions in Hsp90, as indicated in our experiments. Despite the overall conformational dynamics in Hsp90 extends the millisecond timescales, atomistic MD simulations performed on few hundred nanoseconds timescales, can nevertheless also provide important information about the underlying dynamics of the system[41,42]. Moreover, our molecular simulations on three independent Hsp90 mutant support an overall similar picture, suggesting that the overall results are robust and consistent with our experimental findings.

To support our findings and to consider an involvement of the so called "Src-loop" (aa 324–340) in the local interactions, which was found to affect v-Src maturation[15,32] and lies close to W300, we generated two additional mutants, K294A and K325A. Breaking the cation–π interaction in the K294A mutant (Supplementary Fig. 14), could explain why this interaction proposed by our MD simulations, indeed plays an important role in the GR-LBD-Hsp90 interplay. The K325A mutant in the Src-loop was chosen as we could detect strong CSPs for K325 in the W300A background in our NMR experiments (Supplementary Fig. 1, Supplementary Table 1). As K325 lies also within a similar distance from W300 as K294, we speculated that if the Src-loop was involved in the local interactions of W300, this might be a promising candidate. Interestingly, K294A did not show a significant inhibition in its ATPase activity by the GR-LBD implying that the proposed the cation–π interaction is directly involved in the inhibitory mechanism (Fig. 5c). Surprisingly, the K294A mutant did not show significant or strong defects concerning GR-LBD binding independent of the added nucleotides (Fig. 5d). These results explain the regulatory role of W300 and the cation–π interaction to K294 in greater detail. W300 is required to present a conformation that binds the GR-LBD with high affinity and via the cation–π interaction with K294 the inhibitory information of GR-LBD association is transmitted to Hsp90-N. In contrast to K294A, the K325A mutant behaved wt-like. The ATPase activity of the mutant was significantly decreased by the presence of the GR-LBD (Fig. 5c) and GR-LBD binding to the mutant remained unchanged (Fig. 5d) indicating that in the interplay of the GR-LBD with Hsp90, the Src-loop might play a secondary role. Nevertheless, it cannot be ruled out that other residues might be involved or that the loop plays a more critical role in the handling of other clients, especially kinases.

## Discussion

Understanding the molecular mechanism of the Hsp90 chaperone machinery requires detailed insight into the coupling of conformational changes to ATP hydrolysis and client binding. Structural transitions in Hsp90 occur in a highly coordinated manner. This relies on the transfer of structural information over long distances and across domains. Some elements contributing to this process are known, such as the rearrangements in the N-terminal domains in response to ATP binding which eventually lead to their dimerization via β-strand swapping[14,15,43] or

residues in Hsp90-C responding to post-translational modifications[24]. For Hsp90-M, regulatory positions or switch points that affect the conformational equilibrium of Hsp90 or Hsp90's function concerning ATPase activity, client/co-chaperone binding or client maturation are, however, ill-defined. Here, we focused our attention on a conserved tryptophan residue, W300, located in an unstructured loop on the surface of the domain. We find that this residue functions as a switch point in Hsp90-M, important for specific conformational changes associated with client processing. Mutational analysis had shown that exchange of W300 negatively affects yeast viability and GR maturation in vivo and it has been speculated that this residue is also involved in kinase maturation[31,32,35]. Our analysis of various mutants revealed that W300 is important for conformational changes associated with the formation of a semi-closed structure competent for GR-LBD processing. Thus, this residue is a key element in the inter-domain communication in Hsp90 and also connects client binding to changes in the Hsp90 cycle. Our detailed in vitro analysis in combination with information on the conformational equilibrium of Hsp90 and mutants by SAXS as well as MD simulations implies that W300 is essential for Hsp90 to adopt a semi-closed conformation that ensures on the one hand high-affinity binding of the GR-LBD and on the other hand susceptibility to the inhibitory influence of the GR-LBD on the conformational cycle. This is achieved by a cation–π interaction which transmits the information of GR-LBD binding and its inhibitory effect from W300 over K294 to Hsp90-N where the effect is established. Additionally, this conformation is important for client transfer from the Hsp70 chaperone system and processing as shown for efficient induction of hormone rebinding to the GR-LBD. The absence of the tryptophan and thus the lacking cation–π interaction results in an altered, more closed Hsp90 conformation, leading to impaired client handling by the Hsp90 chaperone system (Fig. 6a). Thus, this is the first report on a switch point position that is on the one hand directly involved in priming Hsp90 for high-affinity client binding and on the other hand conveys the cycle modulating information of the client protein via long-range coupling effects from Hsp90-M to Hsp90-N resulting in inhibition of the ATPase activity. This broadens our knowledge of Hsp90 as it resolves the molecular details of how the chaperone is able to transmit cycle-modulating information of client binding from one domain to another.

It is unlikely that W300 is part of the binding site for the GR-LBD. The AUC results show that in the absence of nucleotides, the W300 mutants are able to bind the GR-LBD to a similar extent as the wt protein. Only in the presence of nucleotides, especially those that induce closed conformations, binding to the mutants is reduced whereas it is enhanced for the wt protein indicating that W300 is involved in the formation of a closed conformation that shows high affinity for the GR-LBD. The previously published PRE results support this view. A decrease in signal can only be seen when the spin-label is in close proximity to a residue in the bound state. Further, the spin label is most likely attached to a cysteine outside the interaction site as binding of the GR-LBD to Hsp90-M is not impaired by labeling of the position[27]. This suggests that W300 is in close proximity of the binding site but not within. Given that the client binding-site on Hsp90 is large, one would not assume strong effects upon changing one residue at the periphery, especially if the mutation maintains the aromatic character of the side chain such as the phenylalanine and tyrosine mutations. Thus, it is more likely that W300 is required for triggering a specific conformation that is effective in client binding and promoting the formation of a semi-closed conformation which is susceptible to the cycle inhibiting effects of the GR-LBD. The general influence of W300 on the conformational ensemble of Hsp90 is supported by the

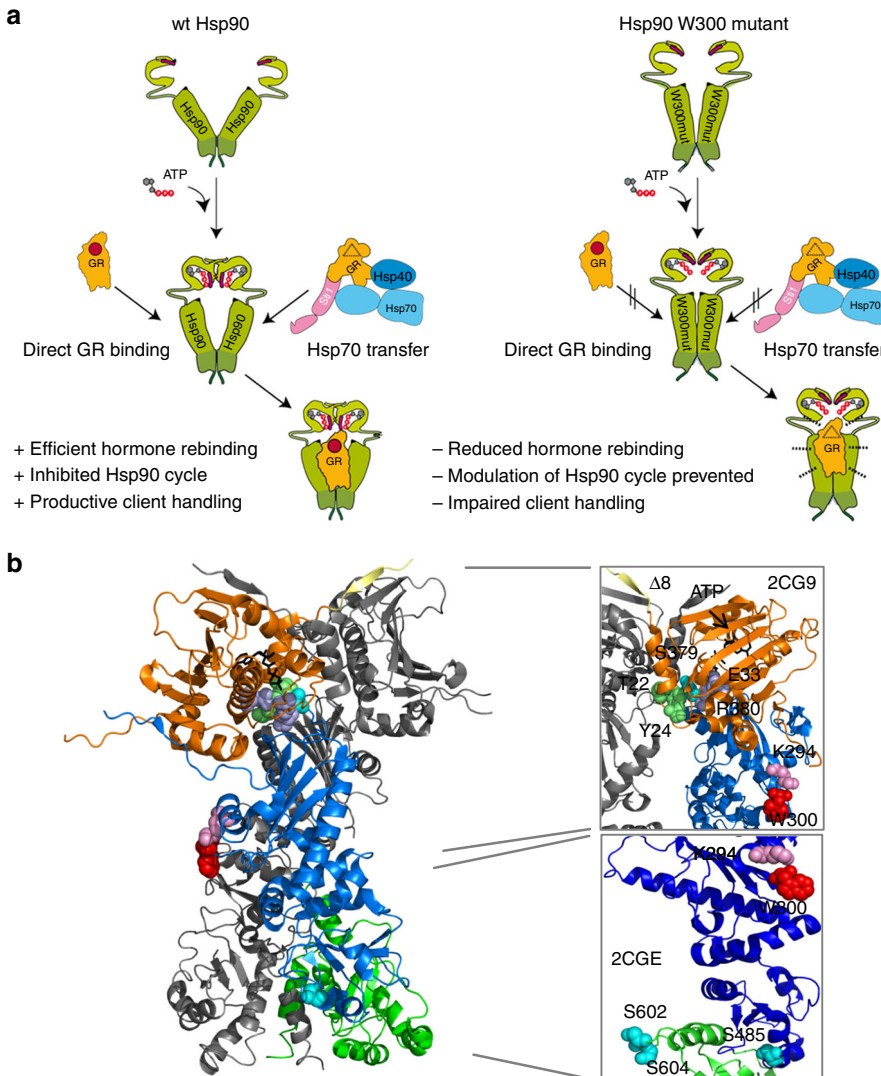

**Fig. 6** Influence of W300 on the conformational cycle of Hsp90. **a** With the help of W300, wt Hsp90 forms a semi-closed conformation induced by the GR that ensures high affinity binding, client transfer from the Hsp70 system, efficient hormone rebinding, establishment of the cycle inhibiting effects of the GR and thereby productive client handling. By forming significantly altered conformations, the Hsp90 W300 mutants show reduced induction of hormone rebinding, an altered association with the GR-LBD and become insensitive to the inhibitory effects of GR on the conformational cycle dynamics as implied by the dotted lines, resulting in impaired client handling of Hsp90. **b** Mapping of the different switch point positions in Hsp90; Switch point positions were mapped on the crystal structure of Hsp90 (PDB ID: 2CG9) and Hsp90-MC (PDB ID: 2CGE). T22 and Y24[20, 46] are colored in lime. S379, S485, S602 and S604[24] are colored in cyan. S602 and S604 are not resolved in the full-length crystal structure and are therefore mapped on the Hsp90-MC structure. R380 and E33 are shown in light purple and the first eight N-terminal residues (Δ8) in yellow[47]. W300 is colored in red and K294 in pink. Boxes show blow-ups from the closed Hsp90 (2CG9) and the 90-MC construct (2CGE)

observation that amino acid residues distant from W300 were affected by the mutations, suggesting that W300 directly influences these regions. Mutants of W300 are still capable of client binding but, in contrast to the wt protein, the ATPase activities of the mutants are unaffected by the presence of the GR-LBD. Interestingly, also the phenylalanine and tyrosine substitutions are defective in respect of alterations in the ATPase activity by the GR-LBD, indicating that a strong π–electron donor is required for induction on this specific conformation in which the ATPase is inhibited by the client protein.

The presented SAXS data implicate that the removal of W300 leads to a change in the conformational equilibrium of Hsp90 with a tendency towards more closed conformations, especially in the case of the W300A and E mutants which negatively affect GR-LBD binding. These mutants reside 10–20% in more closed

conformations than the wt protein in the presence of non-/slowly hydrolysable ATP analogs. This indicates that also moderate changes in the conformational cycle are sufficient to have dramatic effects on client binding and influence of the GR-LBD on Hsp90 as shown by ATPase activity measurements, FRET and AUC binding analysis. The combined results therefore strongly suggest that W300 is a switch point in Hsp90 and thus a control position sensing and directing conformational steps before and after client association. The more closed or generally altered conformations seem to be unfavorable also for the binding of co-chaperones like Aha1, Sgt1 and Sti1[33,34,44]. Since W300 is located close to the binding sites of Aha1 and Sti1 in Hsp90-M[28,44] local changes in the interaction sites might be also important. Other clients like kinases which are also affected by the mutation, might exert similar influences[31]. In this context, the EM structure of the

Hsp90-Cdc37-Cdk4 complex further supports the idea of an involvement of W300 in kinase chaperoning as it lies close to the kinase binding site[45].

Several switch point positions have been described in yeast Hsp90 whose modification affects the conformational cycle or client binding and maturation (Fig. 6b). The bulk of these switch points are phosphorylation sites (T22, Y24, S379, S485, S602, S604) in agreement with an intensive regulation of Hsp90 by kinases and phosphatases[20,24,46]. Changes in the charge pattern due to phosphorylation or mutation differentially affect N-terminal dimerization and ATPase activity, resulting in decreased client maturation[24,46]. Recently, residues and structural motifs were identified that regulate Hsp90 function by affecting the timing of conformational changes[47]. Further, two different classes of regulation points have been discovered which either activate or inhibit the conformational dynamics of the chaperone[26]. Most of the known switch points cluster in Hsp90-N and Hsp90-C or at sub-domain interfaces like the N–M interface including the catalytic loop containing R380, which has been shown to be essential for stabilization of a hydrolysis competent state[26,48,49]. Alterations in both the N-domain and the NM-interface affect not only the local environment but also have consequences on the conformational equilibrium of Hsp90[47]. For example, R380A is trapped in the closed-1 state and is deficient in transiting to the hydrolysis competent closed-2 state. In contrast, the Δ8 construct, in which the residues 1–8 are missing, readily adopts the closed-2 state in the presence of ATP but reopening of this mutant could not be observed. Interestingly, the Δ8 mutant showed significantly reduced GR-LBD binding in the presence of ATP. Thus, the timing of conformational transitions seems essential[47]. Phosphorylation and phospho-mimetic mutations of the switch points in the C-domain (S602, S604) or in close proximity to Hsp90-C (S485) exhibit pronounced long-range structural consequences. The S485E mutation located close to Hsp90-C led to decreased N-terminal dimerization, compaction in the presence of AMP–PNP and GR-maturation[24]. The S602E and S604E variants induced structural changes altering inter-domain communication and Aha1 binding, but left GR-maturation unaffected[24].

The positioning of W300 in comparison to the above-mentioned switch points is significantly different. Whereas most of the known switch points are nearby domain interfaces, W300 is highly surface-exposed in Hsp90-M, supporting a unique status of W300 among the conformational switch points (Fig. 6b). In the case of the phosphorylation sites, introduction of negative charges leads to changes in the local interaction pattern resulting in the desired local and/or long-range conformational changes. W300 might work similar by promoting formation of a special conformation by a local interaction. An exciting and new feature of this switch point is that the local interaction is mediated by a cation–π interaction. The question might arise why this is not also established by a salt bridge. The answer to this lies in the surface exposition of the residue. Surrounding solvent ions could mask the charges and weaken the strength of the salt bridge. A cation–π is not susceptible to salt effects and could thus be the method of choice for Hsp90 to transduce conformational information across the protein especially in solvent-exposed regions which are crucial for client interactions. Further, there is a significant difference in the regulatory requirements in comparison to other switch points. The phosphorylation-dependent switch points seem to be transiently favorable for the cell and therefore are dynamically controlled by kinases, phosphatases and differential co-chaperone influence. In contrast, W300 is a switch point which connects certain Hsp90 conformations to client interaction and thus is constantly required for Hsp90 function and not subject to post-translational modifications. Its importance is highlighted by changes in W300 which lead to the increased presence of partially closed conformations thus deregulating the cycle timing resulting in dysfunctional Hsp90 and decreased client interaction and reactivation. Altogether, the identification of W300 as a switch point integrates the M-domain, for which switch points had not been identified yet, in the network of conformational regulation. In this regard, W300 is a constantly acting switch point which governs the accessibility of an important client-competent conformation at a specific time point during the cycle.

Further constantly active switch regions, but responding to nucleotide binding are β-sheet 1 (aa 1–8) and R380. On the other hand there are the temporarily active PTM-dependent switch points. Within these two classes there might be further sub-divisions depending on the positions and on the aspects modulated by the switch point such as client-handling, co-chaperone association, effects on the conformational equilibrium and influence on the ATPase activity. It will be interesting to see how newly emerging switch points fit this scheme.

## Methods

**Protein expression and purification**. Hsp90/-mutants, human Hsp90β/-mutants, Hsp40 (yeast homolog Ydj-1), human Hsp70, Hop and p23 were expressed and purified from *E. coli*. Human GR-LBD were expressed at 18 °C overnight and purified according to Seitz et al., with minor modifications[50]. For NMR experiments, uniformly labeled samples were isotopically enriched with $^{15}$N. For a more detailed description of the expression and purification protocols, please see Supplementary Methods in the Supplementary Information. Primer sequences are shown in Supplementary Table 4.

**ATPase measurements**. ATPase assays were performed utilizing an ATP-regenerating system[51]. Assays were measured at 30 °C in 40 mM HEPES, 150 mM KCl, 5 mM MgCl$_2$, pH 7.5 (standard buffer) and 2 mM ATP, supplemented with 50 μM DEX using a Varian Cary50 Bio UV-Vis spectrophotometer (Varian Inc., Palo Alto, USA). The GR-LBD was pre-incubated with 3 μM yeast wt or mutant Hsp90 for 10 min. Background ATPase activity was determined by the addition of 50 μM of the Hsp90 inhibitor Radicicol.

**Fluorescence anisotropy**. Fluorescence anisotropy measurements were performed to follow the binding of 50 nM fluorescently labeled Dexamethasone (F-DEX, Sigma-Aldrich, St. Louis, USA) to the GR-LBD in the absence and presence of various chaperone mixtures. For a more detailed description of the experiments, please see the Supplementary Methods in the Supplementary Information. Measurements were conducted on a Jasco Fluorescence Spectrometer FP-8500 equipped with polarizers (Jasco, Groß-Umstadt, Germany). Experiments were performed in 30 mM HEPES, 150 mM KCl, 5 mM MgCl$_2$, pH 7.5 and 5 mM ATP.

**FRET measurements**. Yeast Hsp90 was labeled with donor and acceptor dyes (ATTO-488 and ATTO-550, respectively; ATTO-TEC, Siegen, Germany) at an engineered cysteine residue (C61) in Hsp90-N. FRET measurements were performed in a Jasco Fluorescence Spectrometer FP-8600 (Jasco, Groß-Umstadt, Germany) at 30 °C in standard buffer, supplemented with 50 μM DEX. 200 nM donor-labeled and 200 nM acceptor-labeled yeast Hsp90 were mixed, incubated with increasing concentrations of GR-LBD and the closing reaction was followed after addition of 2 mM ATPγS (Roche, Manheim, Germany). The apparent half-life of the reaction was determined using an exponential decay function. For chase experiments, 4 μM unmodified yeast Hsp90 was added to disrupt the FRET complex.

**Nuclear magnetic resonance NMR experiments**. NMR spectra were recorded using a Bruker AV800, 900, and 950 spectrometers (Bruker Topspin 3.2, Bruker, Billerica, USA) at 25 °C. Unless mentioned otherwise, the NMR buffer used was 20 mM sodium phosphate, 100 mM NaCl, 5 mM EDTA, 0.2% NaN$_3$, pH 6.5 (NMR buffer) and a protein concentration of 600 μM.
Chemical shift assignments are already available for both Hsp90-N domain (aa 1–210) and M-domain (aa 277–527)[27,52,53]. Chemical shift perturbations (CSP) were based on 2D $^1$H,$^{15}$N water-flip-back HSQC correlation experiments comparing the isotopically labeled component of interest in the presence or absence of the binding protein component at natural abundance and calculated as:

$$\Delta\delta_{\mathrm{N-H}} = \sqrt{\left(\Delta\delta_{\mathrm{H}}^{1} * 10\right)^{2} + \left(\Delta\delta_{\mathrm{N}}^{15}\right)^{2}} \qquad (1)$$

**Analytical ultracentrifugation AUC**. For Hsp90 interaction studies, the GR-LBD was randomly labeled with ATTO-488 (ATTO-TEC) on cysteine residues as recommended by the manufacturer. Analytical ultracentrifugation measurements were conducted with a ProteomLab Beckman XL-A centrifuge (Beckman Coulter, Brea, California) equipped with an AVIV fluorescence detection system (Aviv Inc., Lakewood, USA) using 400 nM labeled GR-LBD and 3 µM of the unlabeled components of interest unless noted otherwise. 20 mM HEPES, 20 mM KCl, 5 mM MgCl$_2$, 5 mM DTT pH 7.5 supplemented with 50 µM DEX was used as measurement buffer. Nucleotides (ADP, ATP, ATPγS and AMP–PNP) (Roche, Mannheim, Germany) were added at 2 mM. Data analysis was performed using SedView, SedFit and Origin 8.6[54,55].

**Thermal stability assay**. Thermal unfolding assays were performed to probe the thermal stability of wt Hsp90 and mutants. The assay uses the environment-sensitive fluorescent dye Sypro Orange (Thermo Fischer, Waltham, USA). The dye shows an increase in fluorescence when bound to hydrophobic surfaces exposed during the unfolding of proteins[56]. 0.2 mg/ml protein was mixed with 2 µl 1:1000 (v/v) dilution of Sypro Orange (5000×) to a final volume of 20 µl in standard buffer. The TSA assay was performed in an Agilent Mx3000P QPCR-System (Agilent Technologies Inc., Santa Clara, USA) using a stepwise temperature increase program from 25 to 90.5 °C (131 cycles, 0.5 °C per cycle). The excitation wavelength was set to 470 nm and emission was detected at 570 nm. The melting temperature was determined using the Boltzmann fit function.

**5 fluoro orotic acid FOA shuffling of essential genes**. To test the viability of yeast and human Hsp90 mutants in vivo, a plasmid shuffling assay was performed as described by Nathan and Lindquist[36]. In brief, the *S. cerevisiae* ΔPCLDα strain was used that contains a knock-out of both genomic copies of *hsp90* (*hsp82* and *hsc82*) with a leucine selection marker. The strain carries an URA-selected pKAT6 plasmid coding for *hsp90* constitutively expressed under the control of the glycerinaldehyde-3-phosphate dehydrogenase gene promoter (GPD promoter) to rescue lethality. The URA selection marker enables selection for cells that have lost the wt *hsp90* plasmid in 5-FOA medium. The cells surviving the shuffling were tested for loss of the URA plasmid by plating on selection medium lacking URA. The *hsp90* wt, *hsp90* W300 mutants and human *hsp90ß* genes were expressed from the p413-GPD plasmid containing a HIS selection marker.

**GR activity assay in *S cerevisiae***. The GR activity assay in *S. cerevisiae* was performed to test functionality of yeast Hsp90 mutants in vivo[36]. For this purpose, yeast (ΔPCLDα) were transformed with a reporter plasmid (p2A/GRGZ), carrying a GPD-controlled rat GR gene and a ß-galactosidase gene under control of a Glucocorticoid response element (GRE), and the p413 vector coding for wt *hsp90* or mutants under control of the GPD-promoter. After plasmid shuffling, single clones were picked and grown overnight at 30 °C to stationary phase. The next day, cells were diluted to an OD$_{600}$ of 0.3, induced with 10 µM Deoxycorticosterone (DOC) and incubated for 6–8 h at 30 °C. After induction, OD$_{600}$ was determined and 50 µl of cells were transferred to a 96-well plate and harvested at 4500 rpm for 5 min. Supernatant was discarded, the pellet resuspended in 20 µl SDS-lysis buffer and incubated for 15 min at RT with mixing. After cell lysis, 50 µl 4 mg/ml 2-nitrophenyl ß-D-galactopyranosid (OPNG) in Z buffer were added and ß-galactosidase activity was monitored at 420 nm for 30 min in a Tecan Sunrise plate reader. The absolute activities were calculated by determining the slopes of the reaction and normalized by OD$_{600}$.

**Small angle X-ray scattering SAXS**. Samples were measured on an in-house Anton Paar SAXSess mc2 SAXS instrument equipped with a Kratky camera, a sealed X-ray tube source and a two-dimensional PI SCX:4300 Princeton Instruments CCD detector. The scattering patterns were recorded with a 180-min exposure time (1080 frames, each 10 s). All SAXS data was analyzed with ATSAS (version 2.5). The data was processed with the SAXSQuant software (version 3.9). The forward scattering, $I(0)$, $R_g$, $D_{max}$, desmearing and the inter-atomic distance distribution functions, $P(R)$, were computed with the program GNOM[57]. To calculate the $P(R)$s of mixtures of 'closed' and 'open' Hsp90 conformations, the experimental $P(R)$ of 'open' Hsp90 and the $P(R)$ back-calculated for the 'closed' form of Hsp90 complexes (PDB 2CG9), using the program Crysol were normalized and combined at different ratios[58].

**Atomistic MD simulations**. Simulations were conducted based on full-length yeast Hsp90 dimer with ATP-Mg$^{2+}$ in the closed state, constructed based on the X-ray structure obtained from the Protein Data Bank (PDB ID: 2CG9). Missing loops were modeled using MODELLER[59]. Each model was solvated in a TIP3P water box with 100 mM NaCl concentration with dimension of ca. 176 × 176 × 115 Å$^3$. The molecular systems comprised ca. 300,000 atoms, and were simulated in an NPT ensemble at $T = 310$ K and $p = 101.3$ kPa. The W300 was mutated in silico to Glu, Ala, and Lys, for which independent MD simulations were initiated. All simulations were performed using NAMD 2.9[60] for 200 ns, after 10,000 steps of conjugant gradient minimization, by using an integration time-step of 2 fs using the CHARMM27 force field[61]. The simulations were performed with periodic boundary conditions and treating the long-range electrostatics with the Particle

Mesh Ewald approach using a 1 Å grid size, with a 12 Å switch-distance between near and far-range interactions. Visual MD was used for analysis of both Hsp90 monomers[62]. The Hsp90 reach a dynamic equilibrium during the first 50 ns suggested by stabilization of the root-mean-square-deviation (Supplementary Fig. 14). For the dynamical networks analysis, Hsp90 was coarse grained into edges and nodes based on the last 150 ns MD trajectories. Amino acid residues and Mg$^{2+}$ were each represented by a single node, while the adenine ring, the ribose, and triphosphate unit of ATP, were each represented by separate nodes. Edges were defined as a node contact within 4.5 Å for at least 75% of the analyzed trajectory. The shortest force propagation pathway between nodes in the network was calculated using the Floyd-Warshall algorithm, and the dynamic network analysis was performed using the Carma software[63]. Solvation free energy of the ATP–Mg$^{2+}$ complex in the active site of Hsp90 was calculated along the trajectory by Poisson-Boltzmann continuum electrostatic calculations using the Solinprot module of MEAD[64]. Comparison between chemical shift perturbations obtained from NMR and MD data was made by computing RMSF for the wt and W300A Hsp90 simulations. The RMSF for residue $i$ was computed as, $RMSF_i = 1/N \left[ \Sigma_{ti=1}^{N} (r_i(t_i) - \langle r_i \rangle)^2 \right]^{1/2}$, where $r_i(t_i)$ and $\langle r_i \rangle$ are residue positions at timepoint $t_i$ and average residue positions, respectively. The radius of gyration was computed as $R_g = (\Sigma_i r_i^2 m_i)^2 / \Sigma_i m_i$, where $r_i$ is the position of residue $i$ with respect to its center of mass of the protein, and $m_i$ is the mass of residue $i$. Protein packing densities (Supplementary Fig. 13) were computed as defined[65].

**Statistical analyses**. Statistical significance was assessed using a two-sample $t$-test with a threshold significance level of 0.05. $P$-values above or equal 0.05 were classified as not significant (ns), between 0.05 and 0.01 as significant (*), between 0.01 and 0.001 as very significant (**), between 0.001 and 0.0001 as extremely significant (***) and below 0.0001 also as extremely significant (****).

**Data availability**. The datasets generated and analyzed in this study are available from the corresponding author on reasonable request.

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

## Acknowledgements

We thank M. Biebl for practical assistance, M. Göbl for SAXS measurements, M. Mühlbauer and S. Mader for help in MD analysis, and A. Lopez for reading the manuscript. This work was supported by The Deutsche Forschungsgemeinschaft (grants SFB 1035 A3/B12 to J.B., M.S. and V.R.I.K.) and the Center for Integrated Protein Science Munich (CIPSM). D.A.R. was supported by a fellowship of the "Fonds der Chemischen Industrie". T.M. was supported by the Integrative Metabolism Research Center Graz, the Austrian infrastructure program 2016/2017, BioTechMed/Graz, Omics Center Graz, President's International Fellowship Initiative of CAS (No. 2015VBB045), the National Natural Science Foundation of China (No. 31450110423), and the Austrian Science Fund (FWF: P28854 and W1226-B18). Computer resources were provided by the Gauss Centre for Supercomputing/Leibniz Supercomputing Centre (grant: pr84pa and pr53po).

## Author contributions

D.A.R. performed and analyzed all experiments. L.F. and M.S. performed and analyzed NMR experiments. SAXS measurements were performed and analyzed by T.M., Q.L. and

V.R.I.K performed MD simulations. D.A.R. and J.B. designed, and J.B. supervised the project. The manuscript was written by D.A.R., T.M., V.R.I.K. and J.B.

## Additional information

**Competing interests:** The authors declare no competing interests.

