## [Peer Review File · Nature Communications]

Reviewers' comments:

Reviewer #1 (Remarks to the Author):

This manuscript by Rutz et al. identifies the highly conserved W300 (in yeast Hsp82) aromatic amino acid as a middle domain switch point in Hsp90 that appears to be sensitive to binding of the nuclear receptor client GR. This is an important finding, as switch points in the middle domain of the chaperone are scarce, although mutation of this residue to a non-aromatic amino acid has been known to affect Hsp90 function for some time.

Comments:

1. Although the authors report that GR-LBD binding inhibits wt human Hsp90beta ATPase activity, an older paper from the Jackson lab (J. Mol. Biol. 315: 787, 2002) reported just the opposite observation -namely that GR-LBD stimulated the weak activity of human Hsp90beta by nearly 200-fold. Can the authors comment on these discrepant findings?
2. On page 6, the conclusion of the top paragraph is that W300 is not directly involved in client binding, but instead plays a regulatory role in client interaction. Yet in the last paragraph on page 6 (next section), the authors again present a direct involvement of W300 with the binding of GR-LBD as one of two possibilities to explain the data in Fig. 2A and Table 1. Since they just ruled out this possibility in the section above, I find this to be unnecessarily confusing. The authors should re-write the sentence that begins "There are two possible explanation..." (5th line from bottom of page) to acknowledge that they already discounted one of the explanations a few paragraphs earlier.
3. On page 7, the authors state that W300A and W300E display longer half-lives for formation of heterodimers (and in Fig. 2A have 2-fold increased ATPase activity), while the GR-LBD almost doubles the half-life of heterodimer formation, while inhibiting ATPase activity by 50% (Fig. 2A). Thus, there seems to be no correlation between half-life of heterodimer formation and rate of ATP hydrolysis. Is this correct or am I misinterpreting their data?
4. My major concern is that in all of the bar graphs shown, no tests for statistical significance between groups are reported, even though in the description of the data in the Results, the authors imply numerous instances where differences are important. At least in these cases, an indication of statistical significance should be shown in the figure legends.

Reviewer #2 (Remarks to the Author):

In the paper "A switch point in the molecular chaperone Hsp90 responding to client interaction" Bucher and coworkers have characterized functions of a critical regulatory site Trp300 in the Hsp90-MD using a battery of structural and biochemical approaches. By synergistically employing *in vivo*, *in vitro* and computational methods, the authors studied the effect of multiple mutations in W300 on chaperone structure and binding, thereby probing a general regulatory function of this site in the context of the Hsp90-ATPase cycle and client binding. To test the influence of W300 mutations on the conformational cycle of Hsp90, the authors used FRET system in which donor- and acceptor-labeled Hsp90 dimers allowed to monitor large changes in the Hsp90 dimer and quantify the effects mutations on the conformational cycle of Hsp90 and co-chaperones/client binding. The manuscript presents an interesting and important investigation of Hsp90 regulation and the authors should be commended for taking on this challenge. However, the paper has some major deficiencies that should be carefully considered to make findings of this work more substantiated and complete.

I believe that the following major issues should be considered in that regard:

Major Points

1) The authors have outlined the current state of affairs in understanding switch point positions in Hsp90 that include multiple phosphorylation sites (T22, Y24, S379, S485, S602, S604) in the N-terminal and C-terminal domains. In particular, Buchner and colleagues have previously identified A577 in yeast Hsp90 as an important regulatory site that may control the dynamic equilibrium of Hsp90 and nucleotide-dependent N-terminal dimerization. In my view, the authors should address and clarify functional cross-talk and allosteric coupling between W300 and other regulatory switch sites. Although the results of experiments have unequivocally determined the unique role of W300 in regulation, the molecular basis of these effects is not entirely clear. From structural and dynamic perspective, W300 mutations in the solvent-exposed loop of the Hsp90-M could alter not only client binding through local interactions, but also reorganize amphipathic structure around W300 and change structural arrangement of the entire Src-loop. It would be important to understand how studied mutations affect global conformational changes in the loop and the inter-domain interfaces.

2) Although the authors have shown that mutations of conserved W277 do not have significant effect on Hsp90 regulation, I don't see how it proves that W300 acts "solo" as a single switch. W277 is located in the core, away from W300 and Src-loop, and may be simply irrelevant for allosteric cross-talk in the Hsp90 dimer. On the other hand, mutations in the 329-FDLF-332 region, which is on the other end of Src-loop, are known to affect client binding and regulation (Meyer, Mol. Cell, 2003). It would be interesting to understand the synergistic effect of mutations and cooperativity in W300 and 329-FDLF-332 region in order to make a judgement concerning regulatory switching mechanism. It is not inconceivable to assume that W300 acts as a trigger/initiator of regulatory changes and prompts global changes of the Src-loop structure that, in turn, may activate inter-domain switch points and rewire allosteric communication in the Hsp90 dimer. The authors should consider conducting some additional experiments to address these possibilities.

3) SAXS measurements performed with W300A, W300E and W300K are very interesting but were not sufficiently discussed to better understand structural differences between mutants forms. The central finding here is the fact that the wild type protein resides to 40-50 % in the closed conformation, whereas W300A and W300E are 60-70 % in the closed conformation. These data suggest some modulation of the conformational cycle, but cannot serve as a definitive confirmation of W300 as a switch. In general, the authors should address conceptual meaning and differences between modulation of the Hsp90 cycle and regulatory switching.

4) The authors also conducted MD simulations of WT Hsp90 and W300 mutants to understand molecular basis of the switching mechanism. I found quite interesting that in the course of short 200 ns simulations, they observed signs of structural stabilization of the mutant structures with more compact global structure with increased interactions between the monomers. The dynamic network analysis revealed local coupling between W300 and K294 that may force propagation pathways between the residue 300 (W/A/K/E) and the Hsp90-N domain. Although this is an interesting and plausible rationale, the observed local effects may be present on short time scale (200 ns) and longer simulations are required to prove this mechanism. In addition, the analysis of propagation pathways should be discussed in the context of ensemble of pathways. The question that needs to be addressed is whether the proposed pathway dominated the dynamic ensemble of communication routes between W300 and Hsp90-NTD.

5) It was not obvious from this study, whether the quality and length of MD simulations (200 ns is relatively short for these flexible systems) would have an impact on the results. Are computational predictions sensitive to fluctuations of MD trajectories, or perhaps more coarse-grained elastic network models would have been equally robust and perhaps more appropriate?

6) In discussion, the authors made a good point by comparing the phosphorylation-dependent

switch points and W300, suggesting that W300 is a switch point, which can be sensed by clients, and is constantly required. It would be useful to have a more substantive comparative analysis of regulatory switches and formulate structural and mechanistic rules for determining switch points (or modulation points) in Hsp90. In general, it would be beneficial for Nature readers to learn about general trends and patterns that can characterize regulatory sites that control multiple functions of the Hsp90 chaperone

7) The study lacks sufficient statistical analyses, as there is no information provided about the number of measurement repeats, errors, or statistical significance of the computational results. The authors should also consider to further experimentally and computationally support the main conclusion about conformational switching.

Reviewer #3 (Remarks to the Author):

This manuscript reports the impact of mutating residue W300 of Hsp90 on the protein conformation and activity. The idea is to illustrate the role of the Hsp90 middle-domain in transmitting information from the N-terminal domain to the C-terminal domain. Experiments are performed on yeast and human Hsp90. In yeast, it is shown that W300 is essential for cell viability and glucocorticoid receptor (a model for Hsp90 client) maturation. W300 is close to the glucocorticoid receptor binding site but is not directly involved in binding. W300 mutants do not anymore show an inhibited ATPase activity due to glucocorticoid receptor binding.

At this point the question is: what is the structural explanation for this observation? FRET experiments reveal that mutations of W300 impact the global conformation of Hsp90 dimers, which can be either more open or close depending on the mutation. And in the mutants no conformational difference is observed between the Hsp90 dimer free or bound to the glucocorticoid receptor. The mutated Hsp90 can still adopt a closed conformation in the presence of ATP γ S. And in presence of the glucocorticoid receptor, the closing kinetics is not anymore slowed down, and the reopening is not anymore inhibited. In yeast expressing human Hsp90, tryptophan substitution also impacts cell viability and sensitivity of Hsp90 ATPase activity to the presence of the client glucocorticoid receptor. The presence of mutated Hsp90 is less efficient at inducing glucocorticoid receptor hormone binding. AUC experiments reveal that mutated Hsp90 binds to the client glucocorticoid receptor, however the increase of receptor binding due to ATP γ S is not observed. So here the authors go back to their structural study and show using SAXS that Hsp90 mutants W300A and W300E are characterized by a conformation closer to the close conformation than the WT Hsp90 in the presence of ATP γ S. Molecular dynamics simulation suggests a mechanism for the conformational changes due to mutations at position 300.

The manuscript is not easy to follow for non Hsp90-experts. When the study goes back from the human to the yeast system is not clearly indicated. The link between FRET results and Hsp90 global conformation is just indicated as to be found in publications (17) and (27) without any further details.

Also, why are FRET and SAXS data not presented one after the other as they lead to overlapping conclusions? Wouldn't it be easier to follow? Why the AUC data are not displayed following the FRET and SAXS data to support a global conformational change, WT receptor binding, and then a defect in binding in the presence of ATP γ S?

Could the raw SAXS data be provided as well as the Guinier plot to validate the SAXS analysis? Similarly can the authors show that the P(r) curves correspond to SAXS curves that fit to their experimental curves? Can a table consistent with the guidelines for modelling of SAXS data be provided, as defined in Jacques et al. Acta Crystallographica Section D 2012, 68:620-626? Moreover, the 3D structure displayed in Figure S1 results from a modeling from different X-ray

structures. It is written in the beginning of the results: W300 “is located close to the GR client binding site, in a strongly surface-exposed unstructured loop”. Could it be accurately written from which experimental 3D structure analysis the position of W300 (and the fact that it makes a cation- π interaction with a lysine from the previous helix) is known?

Reviewer #4 (Remarks to the Author):

The authors identify a residue, W300, in the middle domain of Hsp90 that is important for communication between domains and also for client proteins to affect the Hsp90 cycle. This residue is involved in conformational changes associated with client binding elsewhere on Hsp90. The manuscript is well written and the experiments support the conclusions suggested. The significance of how this study improves our understanding of the Hsp90 cycle is not clear or is not clearly presented.

While the paper shows the importance of Hsp82 residue W300, it is not clearly described why this is a “new type of switch point” (in the abstract). It is also not clear how many types of switch there are. Is this a switch of the on/off type? Would it be better described as a residue important for the regulation or modulation (more like a rheostat or dimmer) of the Hsp90 cycle because of its ability to transduce conformational changes over distances following client binding?

Although the location of the W300 residue in the semi-closed conformation is discussed throughout the paper, the location of the W300 residue in the apo and closed/twisted conformations of Hsp90 is not addressed. The significance or lack thereof of mutations in W300 in these other conformations should also be addressed. Since Hsp90 transitions through multiple conformation in the course of the cycle, considering the implications of these other conformations seems important. For example, W300 could be important for some other function, such as stabilizing the open conformation, which could potentially explain why elevated ATPase activity is seen and a more compact conformation for some of the W300 mutants.

Comments:

1. The abstract could be improved since the significance of the paper is not clearly conveyed.
2. The description of the purification of the proteins is not included or was impossible to find. It is important to know if yeast Hsp82 and human Hsp90 were isolated from yeast, since PTMs that are also important for transducing conformation changes which may or may not be present depending on the source of the protein.
3. For Fig. 1B and S4A, it is important to show that the expression level of W300K and W300E is equal to WT in these cells.
4. In Fig. 3D, error bars should be included for the mutants.
5. In 3B, indicate number of repeats.
6. Methods for Fig 3C are not described. Indicate in the legend or methods what components are present, the concentrations of components, the purification of proteins, the time of incubation, etc.
7. In Fig. 4 indicate the number of repeats.
8. Describe simulations in more detail. Indicate the number of trajectories for both WT and mutants and include information such as minimization, solvation details, restraints on the protein during solvation, periodic boundary conditions, how electrostatics were handled, distance cutoffs and update frequencies, etc. Include a supplementary figure indicating that the solvated structure reached equilibrium by 200 ns. Also, consider including a description or formula for RMSF and Rg.
9. It is not clear what is measured in Figure 5B. Please define M-large and M-small.
10. Inclusion of MD simulations complement the experimental work. Perhaps some of the information generated could be used to provide some structural insight into what happens to the network of residues responsible for force propagation in WT vs the mutants. Do these residues form stronger hydrogen bond networks and/or make more contacts with neighboring amino acids

to promote structural rigidity compared to the mutants?

Reviewer #1 (Remarks to the Author):

Comments:

1. *Although the authors report that GR-LBD binding inhibits wt human Hsp90beta ATPase activity, an older paper from the Jackson lab (J. Mol. Biol. 315:787, 2002) reported just the opposite observation -namely that GR-LBD stimulated the weak activity of human Hsp90beta by nearly 200-fold. Can the authors comment on these discrepant findings?*

The reviewer is right about the JMB paper mentioned. However, in an earlier study we already published the inhibitory effect of the GR-LBD on yeast Hsp90s ATPase activity (Lorenz et al., Mol Cell 2014). Also, results from Kirschke and co-coworkers support our findings (Kirschke et al., Cell 2014). In the mentioned paper, a detergent (above the CMC) was used which activates Hsp90, probably due to a crowding effect. In the revised version of the manuscript, we now specifically refer to the previous publications when introducing the assay.

2. *On page 6, the conclusion of the top paragraph is that W300 is not directly involved in client binding, but instead plays a regulatory role in client interaction. Yet in the last paragraph on page 6 (next section), the authors again present a direct involvement of W300 with the binding of GR-LBD as one of two possibilities to explain the data in Fig. 2A and Table 1. Since they just ruled out this possibility in the section above, I find this to be unnecessarily confusing. The authors should re-write the sentence that begins "There are two possible explanation..." (5th line from bottom of page) to acknowledge that they already discounted one of the explanations a few paragraphs earlier.*

We thank the reviewer for noting this. We changed the paragraph accordingly and removed the statement mentioning the two possibilities.

3. *On page 7, the authors state that W300A and W300E display longer half-lives for formation of heterodimers (and in Fig. 2A have 2-fold increased ATPase activity), while the GR-LBD almost doubles the half-life of heterodimer formation, while inhibiting ATPase activity by 50% (Fig. 2A). Thus, there seems to be no correlation between half-life of heterodimer formation and rate of ATP hydrolysis. Is this correct or am I misinterpreting their data?*

There is a correlation between the heterodimer formation and the ATPase activity, but only in the absence of the GR-LBD. In this case W300A/E show longer half-lives, and thus are more closed in the apo-state. It can be speculated that this closed state positively affects the ATPase activity of these mutants. Nevertheless, these mutants do not respond to the presence of the GR-LBD; thus an additional effect on the heterodimer formation is not achieved, in contrast to the wt protein. Wt Hsp90

responds to GR-LBD binding and shows a longer half-life as the presence of the GR-LBD induces a semi-closed conformation. Additionally, the GR-LBD inhibits the Hsp90 ATPase activity indicating that the more closed state induced by the GR-LBD has an inhibitory effect on the ATPase activity in contrast to the state induced by the mutations. We discuss the correlation of the half-life of heterodimer formation and rate of ATP hydrolysis for W300A and E now in the revised manuscript.

- 4. My major concern is that in all of the bar graphs shown, no tests for statistical significance between groups are reported, even though in the description of the data in the Results, the authors imply numerous instances where differences are important. At least in these cases, an indication of statistical significance should be shown in the figure legends.*

As suggested by the reviewer, we now include information on statistical significance where appropriate.

Reviewer #2 (Remarks to the Author):

Comments:

1. *The authors have outlined the current state of affairs in understanding switch point positions in Hsp90 that include multiple phosphorylation sites (T22, Y24, S379, S485, S602, S604) in the N-terminal and C-terminal domains. In particular, Buchner and colleagues have previously identified A577 in yeast Hsp90 as an important regulatory site that may control the dynamic equilibrium of Hsp90 and nucleotide-dependent N-terminal dimerization. In my view, the authors should address and clarify functional cross-talk and allosteric coupling between W300 and other regulatory switch sites. Although the results of experiments have unequivocally determined the unique role of W300 in regulation, the molecular basis of these effects is not entirely clear. From structural and dynamic perspective, W300 mutations in the solvent-exposed loop of the Hsp90-M could alter not only client binding through local interactions, but also reorganize amphipathic structure around W300 and change structural arrangement of the entire Src-loop. It would be important to understand how studied mutations affect global conformational changes in the loop and the inter-domain interfaces. Although the authors have shown that mutations of conserved W277 do not have significant effect on Hsp90 regulation, I don't see how it proves that W300 acts "solo" as a single switch. W277 is located in the core, away from W300 and Src-loop, and may be simply irrelevant for allosteric cross-talk in the Hsp90 dimer. On the other hand, mutations in the 329-FDLF-332 region, which is on the other end of Src-loop, are known to affect client binding and regulation (Meyer, Mol. Cell, 2003). It would be interesting to understand the synergistic effect of mutations and cooperativity in W300 and 329-FDLF-332 region in order to make a judgement concerning regulatory switching mechanism. It is not inconceivable to assume that W300 acts as a trigger/initiator of regulatory changes and prompts global changes of the Src-loop structure that, in turn, may activate inter-domain switch points and rewire allosteric communication in the Hsp90 dimer. The authors should consider conducting some additional experiments to address these possibilities.*

We thank the reviewer for these interesting points. W277 is indeed within the structural core. Our intention for incorporating W277 was to emphasize the importance of W300 by underlining that the determined effects are not seen if we mutate another close-by tryptophan. Further, the W277 data shows that changes within the structural core of the Hsp90-M domain are less relevant than changes in a surface-exposed loop, which seems to be counter-intuitive at first glance, but supports the importance of W300. In our revised manuscript, we focused on the molecular basis of the W300 effect in greater detail. Based on the results from the MD simulations, we generated two additional mutants, K294A, which is the lysine that forms cation- π interaction with W300, and K325A located in the so-called Src-loop. In additional ATPase activity measurements and AUC analyses, we found that the cation- π interaction of K294 with W300 is crucial to induce the inhibiting effects of the GR-LBD on the conformational cycle of Hsp90. Mutating K294 to alanine abolished the inhibitory effect of the GR-LBD on Hsp90's ATPase activity. Interestingly, GR

binding to this mutant was not strongly affected under all tested nucleotide conditions in contrast to the W300-mutants which showed severe binding defects especially for the closed conformations. These new data explain the function of the client-sensing switch point W300 in greater detail. When W300 and K294 are present, W300 is important for inducing a conformation that binds the client with high affinity. Via the cation- π interaction with K294 the inhibitory effect of the client is transduced to the N-terminal domain, decelerating ATP hydrolysis as also implicated by the MD simulations. When K294 is absent, client binding can occur normally but the inhibitor information cannot be transmitted via the cation- π interaction. Mutation of W300 is even worse as neither the correct client-binding conformation can be formed, nor the inhibitory information can be transported to Hsp90-N. These new data are included and discussed in the revised manuscript.

As proposed by the reviewer, we also had a look at a residue in the Src-loop, K325 which is in similar distance to W300 as K294 and which experiences strong chemical shift perturbations due to the W300A mutation. For this mutant, we did not detect significant changes in the influence on the ATPase activity and on GR-LBD binding to Hsp90. We conclude from these measurements that the Src-loop might not be strongly involved in the GR interaction with Hsp90 and allosteric communication. Nevertheless, there is still the possibility that W300 affects the residues in the Src-loop and that this has a more pronounced effect on other clients. The data for both new mutants is included in the revised manuscript.

2. *SAXS measurements performed with W300A, W300E and W300K are very interesting but were not sufficiently discussed to better understand structural differences between mutants forms. The central finding here is the fact that the wild type protein resides to 40-50 % in the closed conformation, whereas W300A and W300E are 60-70 % in the closed conformation. These data suggest some modulation of the conformational cycle, but cannot serve as a definitive confirmation of W300 as a switch. In general, the authors should address conceptual meaning and differences between modulation of the Hsp90 cycle and regulatory switching.*

We thank the reviewer for this comment. The reviewer is right that 10-20% change does not sound very strong but the SAXS data presented here should not be seen in isolation. This change in the conformational spectrum has dramatic influences on the binding and handling of the GR-LBD. W300 might not be a classical switch point like the delta8-mutant (Zierer et al., NSMB 2016) which changes the conformational landscape/cycle of Hsp90 dramatically. W300 has a pronounced effect in the context of the client which is, as we are convinced more important. In our modified manuscript we now include a revised discussion of the SAXS data. In the beginning of the discussion we define switch points as positions “that affect the conformational equilibrium of Hsp90 or Hsp90s function concerning ATPase, client/co-chaperone binding or client maturation” and restrain from using the term modulation in the context of switch points as there is no defined difference between switching and modulating.

3. *The authors also conducted MD simulations of WT Hsp90 and W300 mutants to understand molecular basis of the switching mechanism. I found quite interesting that in the course of short 200 ns simulations, they observed signs of structural stabilization of the mutant structures with more compact global structure with increased interactions between the monomers. The dynamic network analysis revealed local coupling between W300 and K294 that may force propagation pathways between the residue 300 (W/A/K/E) and the Hsp90-N domain. Although this is an interesting and plausible rationale, the observed local effects may be present on short time scale (200 ns) and longer simulations are required to prove this mechanism. In addition, the analysis of propagation pathways should be discussed in the context of ensemble of pathways. The question that needs to be addressed is whether the proposed pathway dominated the dynamic ensemble of communication routes between W300 and Hsp90-NTD.*

We thank the reviewer for these suggestions. In addition to the WT-Hsp90, we also studied the W300A, W300E, and W300K mutations, all of which show a similar perturbation in the local residue interaction network. This suggest that the overall results are robust. MD simulations performed for a few hundred nanoseconds, have previously been shown to capture local conformational changes in proteins, particularly if there is an introduction of a perturbation in the system as here (the W300A/E/K mutations). However, we now better emphasize that the molecular simulations presented here should not be considered in isolation, but as part of our efforts to probe structural and mechanistic aspects of W300. To further validate the reviewer`s question regarding the W300-K294 interactions, we mutated also the latter residue experimentally. In the K294A, the inhibitory effect of the GR-LBD on Hsp90's ATPase activity is abolished, supporting the relevance of the simulations. This has now been clarified in the revised text. Longer MD simulations are currently out of the scope of the present work, but the RMSD analyses, now shown in Figure S14, indicate that the models stabilize globally on the simulation timescales. Interactions and dynamics along the putative communication pathway have been analyzed Figure S11, and the text has been revised accordingly.

4. *It was not obvious from this study, whether the quality and length of MD simulations (200 ns is relatively short for these flexible systems) would have an impact on the results. Are computational predictions sensitive to fluctuations of MD trajectories, or perhaps more coarse-grained elastic network models would have been equally robust and perhaps more appropriate?*

RMSD analyses, now shown in Figure S14, indicate that Hsp90 reaches equilibrium on the simulation timescales. Elastic networks can indeed provide ideas of protein dynamics. However, such models are not usually based on the exact amino acid composition but rather the coarse-grained C_{alpha}-atom representation of the protein. It is therefore unlikely that such models could capture the local re-arrangement of the side-chains. Coarse-grained MD simulations can also provide powerful ways to address longer timescales, but such approaches require restraining secondary structure elements as the CG-force field do not model such elements accurately. We therefore decided that classical atomistic MD simulations provide the necessary

feedback required for the current work. Due to the large size of the system, longer simulations are currently out of the scope of the present work. The MD simulations, however, provide an important rationale to explain our experimental findings, and to further predict and verify the cation- π interaction, which we now have further validated experimentally (see above).

- 5. In discussion, the authors made a good point by comparing the phosphorylation-dependent switch points and W300, suggesting that W300 is a switch point, which can be sensed by clients, and is constantly required. It would be useful to have a more substantive comparative analysis of regulatory switches and formulate structural and mechanistic rules for determining switch points (or modulation points) in Hsp90. In general, it would be beneficial for Nature readers to learn about general trends and patterns that can characterize regulatory sites that control multiple functions of the Hsp90 chaperone.*

This is an excellent point and we have tried to better present the concepts that emerge from this and previous studies in the revised version.

- 6. The study lacks sufficient statistical analyses, as there is no information provided about the number of measurement repeats, errors, or statistical significance of the computational results. The authors should also consider to further experimentally and computationally support the main conclusion about conformational switching.*

We thank the reviewer for noticing this. In our revised version of the manuscript, we incorporated information on the number of repeats and statistical significance in the figure legends where appropriate. We now better point out that three independent MD simulations of W300 mutations support a similar picture and suggest that the overall results are robust. Moreover, due to the Hsp90 dimer, each simulation comprises two "W300s". The computationally predict W300 \rightarrow K294 interaction was also experimentally verified (see above), further supporting the significance of the computer simulations. We also discuss in more detail the correlation between MD and NMR.

Reviewer #3 (Remarks to the Author):

Comments:

1. *The manuscript is not easy to follow for non Hsp90-experts. When the study goes back from the human to the yeast system is not clearly indicated. The link between FRET results and Hsp90 global conformation is just indicated as to be found in publications (17) and (27) without any further details.*

In response to the reviewer's comment, we now clearly indicate when we use human Hsp90 or the yeast system. Further, we have now discussed the link between FRET and global conformational changes in more detail.

2. *Also, why are FRET and SAXS data not presented one after the other as they lead to overlapping conclusions? Wouldn't it be easier to follow? Why the AUC data are not displayed following the FRET and SAXS data to support a global conformational change, WT receptor binding, and then a defect in binding in the presence of ATP γ S?*

We thank the reviewer for this comment. First of all, we wanted to draw the reader's attention to the functional defects of the W300 mutants observed *in vitro*. This includes the ATPase, FRET and hormone-binding results. In a next step, we rule out that these deficiencies are based on a general binding defect. As the AUC results showed that only specific nucleotide states are affected indicating that the conformational spectrum might be changed, we have a closer look on this by SAXS. We did not want to split up the AUC binding results as proposed by the reviewer as we think this might cause problems in following the central theme of our manuscript which goes from functional defects to their structural explanation. We mention this central theme as the rationale for the order of experiments now in the revised version of the manuscript.

3. *Could the raw SAXS data be provided as well as the Guinier plot to validate the SAXS analysis? Similarly can the authors show that the $P(r)$ curves correspond to SAXS curves that fit to their experimental curves? Can a table consistent with the guidelines for modelling of SAXS data be provided, as defined in Jacques et al. Acta Crystallographica Section D 2012, 68:620-626?*

We have included the raw SAXS data, including the Guinier plots in supplementary figures S5-S8 and an updated table consistent with the published guidelines in supplementary table S2. The $P(R)$ curves have been calculated using the program GIFT, including a desmearing step of the line collimated SAXS data. Not all parameters listed in the table published by Jacques et al. Acta Crystallographica Section D 2012, 68:620-626 are available using the currently available software GIFT and ATSAS (SAS data analysis with desmearing option for line collimated data). All

raw data, GIFT output files, and the beam profiles have been submitted to SASDB and will be released upon publication of the manuscript.

4. *Moreover, the 3D structure displayed in Figure S1 results from a modeling from different X-ray structures. It is written in the beginning of the results: W300 “is located close to the GR client binding site, in a strongly surface-exposed unstructured loop”. Could it be accurately written from which experimental 3D structure analysis the position of W300 (and the fact that it makes a cation- π interaction with a lysine from the previous helix) is known?*

We thank the reviewer for this comment. The 3D structures shown in Figure S1 are not the product of a modelling approach. As mentioned now in the figure legend, the Hsp90-M domain (res 259-527) from the closed Hsp90 full-length structure (PDB ID: 2CG9) is shown and the intensity changes and CSPs are mapped. The location of W300 in the Hsp90 context was deduced from the crystal structure of full length Hsp90 (PDB ID: 2CG9). The proximity of W300 to the GR-LBD binding site was derived from the mapping of the GR-LBD binding site on Hsp90 from Lorenz et al., *Molecular Cell* 2014. The cation- π interaction was deduced from the dynamic proximity of K294 and W300 in the MD simulations, which forms contacts $< 5-6 \text{ \AA}$, a distance in which 99% of significant cation- π interactions involving tryptophans occur (Gallivan and Dougherty, *PNAS*, 1999). The cation- π interactions and respective distance plot are now shown in Figure S10. The respective section and figure legend have been extended to address these points in more detail.

Reviewer #4 (Remarks to the Author):

Comments:

1. *The authors identify a residue, W300, in the middle domain of Hsp90 that is important for communication between domains and also for client proteins to affect the Hsp90 cycle. This residue is involved in conformational changes associated with client binding elsewhere on Hsp90. The manuscript is well written and the experiments support the conclusions suggested. The significance of how this study improves our understanding of the Hsp90 cycle is not clear or is not clearly presented.*

We thank the reviewer for this comment. To make the significance of our story clearer we insert the following statement in our revised discussion: “Thus, this is the first report on a switch point position that is on the one hand directly involved in priming Hsp90 for high-affinity client binding and on the other hand conveys the cycle modulating information of the client protein via long-range coupling effects from Hsp90-M to Hsp90-N resulting in inhibition of the ATPase activity. This broadens our knowledge of Hsp90 as it resolves the molecular details of how the chaperone is able to transmit cycle-modulating information of client binding from one domain to another.”

2. *While the paper shows the importance of Hsp82 residue W300, it is not clearly described why this is a “new type of switch point” (in the abstract). It is also not clear how many types of switch there are. Is this a switch of the on/off type? Would it be better described as a residue important for the regulation or modulation (more like a rheostat or dimmer) of the Hsp90 cycle because of its ability to transduce conformational changes over distances following client binding?*

In response to the reviewer’s comment, we now explain in the abstract why W300 is a new type of switch point. In the discussion we also mention a variety of switch points that exist in Hsp90 such as PTM-dependent switches and also propose a potential clustering for the different types. In contrast to PTM-driven switch points, W300 is not an on/off switch point. To address this more clearly, we added the following statement to the discussion: “In contrast, W300 is a switch point which connects certain Hsp90 conformations to client interaction and thus is constantly required for Hsp90 function and not subject to post-translational modifications.” The term switch point could certainly be changed to rheostat or dimmer as suggested, but in order to not include an additional term in the field of Hsp90, we would prefer to stick to switch point. This is also beneficial for readers that are not experts in the field, as the differentiation between switch point and dimmer might be confusing. Further, we prefer the term switch point because as soon as the GR-LBD is associated, W300 “switches on” the inhibitory effect on the conformational cycle by transmitting the information on binding via K294 to Hsp90-N.

3. *Although the location of the W300 residue in the semi-closed conformation is discussed throughout the paper, the location of the W300 residue in the apo and closed/twisted conformations of Hsp90 is not addressed.*

The surface-exposed position in the closed/twisted conformation as shown in Figure 1A is now discussed in the first result paragraph. Due to the lack of structural data, we cannot provide information about the exact positioning of W300 in the open apo conformation.

4. *The significance or lack thereof of mutations in W300 in these other conformations should also be addressed. Since Hsp90 transitions through multiple conformation in the course of the cycle, considering the implications of these other conformations seems important. For example, W300 could be important for some other function, such as stabilizing the open conformation, which could potentially explain why elevated ATPase activity is seen and a more compact conformation for some of the W300 mutants.*

We thank the reviewer for this comment. It would be very interesting to understand the role of W300 in each of the conformations Hsp90 adopts during the cycle. We have indications for the role of W300 in the open conformation and closed state as shown by FRET which indicates that the apo-state is already slightly more closed. Nevertheless, the closing kinetics remain unchanged and also the closed state is not dramatically altered in comparison to the wt-protein as seen in the subunit-exchange kinetics. These findings are also supported by the SAXS measurements. Thus, we can rule out that W300 stabilizes the open conformation as this would not fit to our results. Additionally, conformations that are in between open and closed are very difficult to access at present. This would require the development of novel experimental approaches that go beyond the scope of our manuscript.

5. *The abstract could be improved since the significance of the paper is not clearly conveyed.*

As suggested, the abstract has been revised.

6. *The description of the purification of the proteins is not included or was impossible to find. It is important to know if yeast Hsp82 and human Hsp90 were isolated from yeast, since PTMs that are also important for transducing conformation changes which may or may not be present depending on the source of the protein.*

We thank the reviewer for this comment. The purification protocols are now included in the supplementary information. Hsp90 heterologues/mutants were purified from *E. coli*. Therefore, posttranslational modifications are absent. This is also mentioned in the revised manuscript.

7. *For Fig. 1B and S4A, it is important to show that the expression level of W300K and W300E is equal to WT in these cells.*

In response to the reviewer's comment, we incorporated Western-Blot analysis of the viable W300 mutants (A, F, Y and W277A) into the supplemental information (Figure S1A). For the W300E and K mutants which are not viable, the mutant levels cannot be quantified.

8. *In Fig. 3D, error bars should be included for the mutants.*

Has been done.

9. *In 3B, indicate number of repeats.*

The measurements were performed in triplicates. This is indicated in the revised version of the manuscript.

10. *Methods for Fig 3C are not described. Indicate in the legend or methods what components are present, the concentrations of components, the purification of proteins, the time of incubation, etc.*

We thank the reviewer for noticing this. We have now included an extended materials and method section in the supplementary information. The experiments were performed as described by Kirschke et al., Cell 2015; Incubation times are now mentioned in Figure S4D, in the schematic description of the experiment.

11. *In Fig. 4 indicate the number of repeats.*

SAXS measurements are typically carried out as single measurements at different sample concentrations. The corresponding SAXS raw data is shown in supplementary figures S5-S8. The error bar of each data point shown in supplementary figures S5-S8 is derived from automatic averaging the SAXS signal of individual detector pixels and measurement frames (i.e. 540 frames of 10 seconds each in a 90 minute measurement).

12. *Describe simulations in more detail. Indicate the number of trajectories for both WT and mutants and include information such as minimization, solvation details, restraints on the protein during solvation, periodic boundary conditions, how electrostatics were handled, distance cutoffs and update frequencies, etc. Include a supplementary figure indicating that the solvated structure reached equilibrium by 200 ns. Also, consider including a description or formula for RMSF and Rg.*

The simulations details have now been added. Equations for RMSF and Rg are now given in the Methods section. Equilibration measures are shown in the new Figure S14.

13. *It is not clear what is measured in Figure 5B. Please define M-large and M-small.*

M-large and M-small and measured distances have now been defined in Figure 5B.

14. *Inclusion of MD simulations complement the experimental work. Perhaps some of the information generated could be used to provide some structural insight into what happens to the network of residues responsible for force propagation in WT vs the mutants. Do these residues form stronger hydrogen bond networks and/or make more contacts with neighboring amino acids to promote structural rigidity compared to the mutants?*

An analysis of the network dynamics and contacts in the simulation ensemble is now included in the revised manuscript and shown in Figure S11.

REVIEWERS' COMMENTS:

Reviewer #1 (Remarks to the Author):

This revision is a significant improvement over the original submission. The authors have strengthened their study by providing requested clarifications and additional experiments suggested by the reviewers.

Reviewer #2 (Remarks to the Author):

The revision represents a significant improvement and some additional experiments provided much needed clarity and better rationalization of the results.